# Q-Insight: Understanding Image Quality via Visual Reinforcement Learning

**Weiqi Li[1], Xuanyu Zhang[1], Shijie Zhao[2]♦✉, Yabin Zhang[2], Junlin Li[2], Li Zhang[2], Jian Zhang[1]✉**

[1] School of Electronic and Computer Engineering, Peking University

[2] ByteDance Inc.

## Abstract

Image quality assessment (IQA) focuses on the perceptual visual quality of images, playing a crucial role in downstream tasks such as image reconstruction, compression, and generation. The rapid advancement of multi-modal large language models (MLLMs) has significantly broadened the scope of IQA, moving toward comprehensive image quality understanding that incorporates content analysis, degradation perception, and comparison reasoning beyond mere numerical scoring. Previous MLLM-based methods typically either generate numerical scores lacking interpretability or heavily rely on supervised fine-tuning (SFT) using large-scale annotated datasets to provide descriptive assessments, limiting their flexibility and applicability. In this paper, we propose **Q-Insight**, a reinforcement learning-based model built upon group relative policy optimization (GRPO), which demonstrates strong visual reasoning capability for image quality understanding while requiring only a limited amount of rating scores and degradation labels. By jointly optimizing score regression and degradation perception tasks with carefully designed reward functions, our approach effectively exploits their mutual benefits for enhanced performance. Extensive experiments demonstrate that Q-Insight substantially outperforms existing state-of-the-art methods on both score regression and degradation perception tasks, while exhibiting impressive zero-shot generalization and superior comparison reasoning capability. The code and models are available at `https://github.com/bytedance/Q-Insight`.

## 1 Introduction

Image quality assessment (IQA) is a fundamental task in computer vision, critical for optimizing algorithms, enhancing user experiences, and verifying content authenticity across diverse domains, such as image processing [41, 22, 11, 53] and AI-generated content (AIGC) [37, 8]. Traditional IQA methods rely heavily on hand-crafted metrics, either through reference-based comparisons [50] or statistical measures of natural image properties [64]. However, these approaches typically focus on local image characteristics and fail to comprehensively capture global visual quality, limiting their reliability in complex real-world scenarios. More recently, deep learning-based IQA models [47, 19] have emerged, utilizing neural networks to learn hierarchical image representations. Nevertheless, these methods struggle to face significant challenges in out-of-distribution (OOD) generalization.

With recent advances in multi-modal large language models (MLLMs) [23, 1, 48], researchers have begun to leverage these models' extensive world knowledge and perceptual abilities to enhance IQA performance and broaden its applicability [53, 60, 59, 58, 52]. Existing MLLM-based IQA methods generally fall into two categories: score-based methods, such as Q-Align [53] and DeQA-Score [58], and description-based methods, exemplified by DepictQA [60] and DepictQA-Wild [59]. Score-based methods transform discrete tokens into continuous quality scores, thereby improving

---

♦Project Lead. ✉: Corresponding authors, zhaoshijie.0526@bytedance.com, zhangjian.sz@pku.edu.cn.

†This work was supported in part by National Natural Science Foundation of China (No. 62372016).

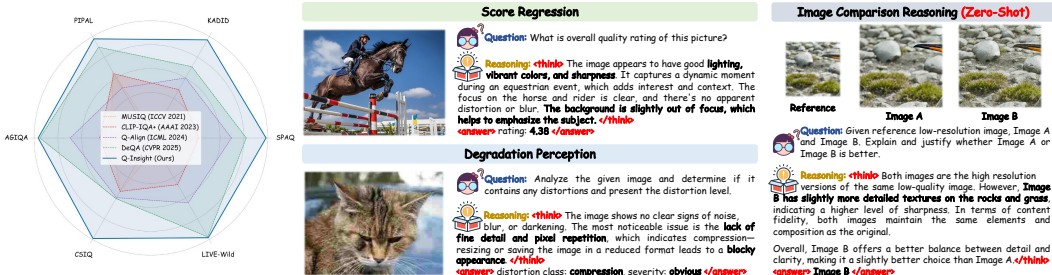

Figure 1: PLCC comparisons between our proposed Q-Insight and existing IQA metrics (left) and three example applications of our Q-Insight (right) are presented. Q-Insight demonstrates significantly improved performance compared to existing methods such as DeQA-Score [58], especially on out-of-domain datasets (e.g., CSIQ [20]). Additionally, Q-Insight effectively supports quality score regression, image degradation perception, and zero-shot image comparison reasoning tasks.

adaptability but typically sacrificing interpretability and neglecting MLLMs' intrinsic reasoning and descriptive capabilities. Meanwhile, **simply regressing a quality score may not be meaningful in certain scenarios**, as image quality scores are subjective, inherently biased, and lack uniform standards across different datasets and content types. For example, when evaluating AIGC-generated data, unusual visual effects and vibrant colors often imply better quality. However, for evaluating super-resolution results, these same features are often considered too painterly, losing the image's authenticity and fidelity. Conversely, description-based methods produce detailed textual explanations of image degradations and comparative assessments, ensuring interpretability yet heavily depending on extensive textual depiction for supervised fine-tuning. Moreover, these models cannot output precise scores, making them unsuitable when used as loss functions or for performing accurate ranking of image quality. Consequently, integrating **numerical scoring** and **descriptive reasoning** into a unified, interpretable MLLM-based IQA framework remains an essential yet unresolved challenge.

In this paper, we move towards a comprehensive understanding of image quality by addressing tasks such as image description, aesthetic and compositional evaluation, degradation perception, and comparative reasoning across images. Rather than teaching the large model "*How to score images*", we aim to inspire it "*How to reason deeply and formulate insightful perspectives on image quality metrics during scoring*". To this end, we resort to Group Relative Policy Optimization (GRPO) [40], a reinforcement learning framework inspired by DeepSeek-R1 [15]. GRPO has recently shown to be highly effective in large language models (LLMs). It uses heuristic reward signals to efficiently guide LLMs in uncovering their intrinsic reasoning capabilities, removing extensive reliance on annotated reasoning chains or additional value models. Recently, researchers have also successfully adapted GRPO to vision-language tasks, including few-shot object detection, reasoning grounding [25], and medical analysis [34]. In the context of image quality understanding, the introduction of GRPO provides at least three distinct advantages: **(1)** no reliance on massive textual training data, **(2)** strong generalization to OOD evaluated images, and **(3)** high diversity in supporting multiple tasks. These benefits align well with our goal of developing a generalized image quality understanding agent.

Specifically, we design **Q-Insight** upon the GRPO framework. In our Q-Insight, we jointly optimize score regression and degradation perception tasks, and carefully design three reward functions: a verifiable score reward for the score regression task, and degradation classification and intensity perception rewards for the degradation perception task. Consequently, Q-Insight effectively exhibits robust reasoning performance using only limited Mean Opinion Scores (MOS) and degradation labels. As shown in Fig. 1, our Q-Insight delivers remarkable performance improvements especially on OOD datasets, while demonstrating comprehensive capabilities across multiple quality assessment and reasoning tasks. For example, it can accurately identify cases where a slightly blurred background, usually regarded as undesirable, effectively helps to emphasize the primary subject of the image. Our empirical investigation reveals that: **(1)** training solely with score labels results in poor perception of image detail degradations (e.g., JPEG compression), while jointly training with the degradation perception task significantly enhances the model's sensitivity to such degradations, and **(2)** score regression and degradation perception tasks are mutually beneficial. Extensive experiments across score regression and degradation perception tasks demonstrate that Q-Insight consistently outperforms existing model-based IQA metrics as well as SFT-driven large language models. Moreover, it exhibits impressive zero-shot generalization on unseen tasks, such as image comparison reasoning, highlighting the robustness and versatility of our method. In summary, our contributions are:

❑ (1) We propose Q-Insight, the first reasoning-style multi-modal large language model specifically designed for comprehensive image quality understanding. Unlike previous methods that depend heavily on detailed textual descriptions for supervised fine-tuning (SFT), our approach achieves superior understanding capability using only limited mean opinion scores or degradation labels.

❑ (2) We introduce a unified framework that jointly optimizes image quality rating and degradation perception, revealing mutual benefits across tasks. Within this framework, we develop three specialized rewards, including verifiable score reward, degradation classification and intensity perception rewards, enabling the GRPO framework to effectively generalize to low-level vision applications.

❑ (3) Extensive experiments across diverse datasets and IQA tasks demonstrate that Q-Insight consistently outperforms existing model-based IQA metrics as well as SFT-driven large language models. Moreover, it exhibits impressive zero-shot generalization on unseen tasks, such as reference-based image comparison reasoning, highlighting the robustness and versatility of our method.

## 2 Related Work

**Score-based IQA methods** include full-reference and no-reference approaches. Full-reference methods [50, 42, 65] assess image quality by comparing distorted images with high-quality references using traditional metrics (e.g., SSIM [50]) or advanced deep-learning-based metrics [3, 4, 10, 9, 13, 36] like LPIPS [66]. Non-reference methods evaluate quality without reference images, shifting from traditional handcrafted statistics [27, 28, 29, 30, 31, 38] to deep-learning-derived quality priors [18, 19, 24, 33, 44, 73, 74, 45, 49]. Recent multi-modal large language model (MLLM)-based methods, such as Q-Align [53] and DeQA-Score [58], leverage MLLMs' knowledge and perceptual abilities to produce scores. However, they sacrifice the intrinsic descriptive capabilities of MLLMs.

**Description-based IQA methods** utilize the foundational knowledge of MLLMs to deliver detailed qualitative assessments and improved interpretability [51, 52, 60, 59, 54, 5, 70, 71, 69]. For instance, Q-Bench [51] and Q-Instruct [52] enhance the low-level perceptual capabilities of MLLMs through specialized datasets and tailored evaluation strategies. Co-Instruct [54] specifically focuses on comparative quality assessments among multiple images. Approaches such as DepictQA [60] and DepictQA-Wild [59] handle both single-image and paired-image evaluations across full-reference and no-reference scenarios. Q-Ground [5] emphasizes a detailed visual quality analysis through visual grounding. However, these methods are highly dependent on extensive textual annotations for supervised fine-tuning, leading to considerable costs in human labor or GPT token consumption.

**Reinforcement learning (RL)** has emerged as an effective strategy to enhance the reasoning performance of LLMs through feedback-driven refinement [7, 43, 40, 55, 57, 17, 68]. Methods like RLHF [32] and RLAIF [2] employ human or AI-generated feedback to refine model behavior. In vision-language tasks, RL has successfully been employed to align model predictions closely with human preferences and reduce hallucinations [46, 61, 62, 72]. Recently, DeepSeek-R1-Zero [15] introduced group relative policy optimization (GRPO) [40], leveraging rule-based rewards to strengthen reasoning capabilities without supervised fine-tuning. Furthermore, Visual-RFT [25] applied GRPO to visual grounding, and Med-R1 [34] adopted GRPO for medical reasoning tasks. R1-VL [63] extends GRPO through step-wise optimization for multi-modal reasoning. Distinctly, our Q-Insight is the first to integrate RL-based strategies into the foundational visual quality understanding model. It jointly trains on multiple tasks and demonstrates mutually beneficial effects among them.

## 3 Methodology

### 3.1 Preliminaries

**Group Relative Policy Optimization (GRPO)** is an innovative reinforcement learning paradigm that has been widely used in models such as DeepSeek R1-Zero [15], achieving excellent results. Unlike Proximal Policy Optimization (PPO) [39], which requires an explicit critic model to evaluate the performance of the policy model, GRPO [40] directly computes the advantage by comparing a group of responses sampled from the policy model, greatly reducing the computational burden. Specifically, given a query $q$, GRPO samples $N$ distinct responses $\{o^{(1)}, o^{(2)}, \ldots, o^{(N)}\}$ from the old policy $\pi_{\theta_{\mathrm{old}}}$. Then, the method performs the corresponding actions and receives the respective rewards $\{r^{(1)}, r^{(2)}, ..., r^{(N)}\}$ according to the task-specific rules. By calculating the mean and standard

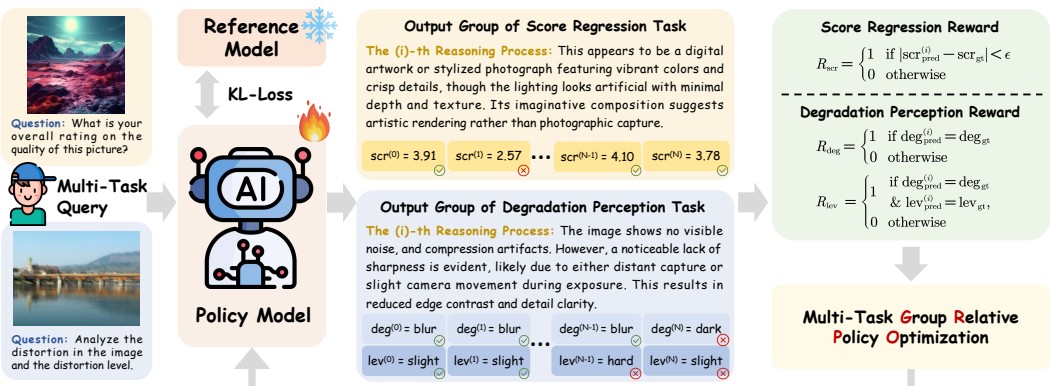

Figure 2: **Overview of the proposed Q-Insight framework.** The policy model receives queries from multiple tasks and generates corresponding groups of responses accompanied by explicit reasoning steps. Task-specific reward functions ($R_{scr}$, $R_{deg}$, and $R_{lev}$) are then applied, and the policy model is subsequently optimized jointly using the multi-task group relative policy optimization algorithm.

deviation of the rewards, the relative advantages of each response can be obtained as follows:

$$\hat{A}^{(i)} = \frac{r^{(i)} - \text{mean}(\{r^{(1)}, r^{(2)} \ldots, r^{(N)}\})}{\text{std}(\{r^{(1)}, r^{(2)} \ldots, r^{(N)}\})}, \tag{1}$$

where $\hat{A}^{(i)}$ represents the normalized relative quality of the $i$-th answer. Overall, GRPO guides the policy model to prioritize higher-quality answers that receive higher reward values within the group. After obtaining the advantage $\hat{A}^{(i)}$, GRPO calculates the ratio of the probabilities of each response under the new policy $\pi_{\theta_{new}}$ and the old policy $\pi_{\theta_{old}}$, denoted as $\rho^{(i)}$. To prevent overly large updates to the model and stabilize training, GRPO restricts the $\rho^{(i)}$ to the range $[1 - \delta, 1 + \delta]$. To further maintain closeness to the reference distribution $\pi_{ref}$, a KL divergence penalty weighted by $\beta$ is adopted. Finally, the optimization objective of GRPO can be formulated as follows:

$$\mathcal{J}(\theta) = \mathbb{E}_{[q \sim Q, o^{(i)} \sim \pi_{\theta_{old}}]} \left\{ \min \left[ \rho^{(i)} \hat{A}^{(i)}, \text{clip}\left(\rho^{(i)}, 1 - \delta, 1 + \delta\right) \hat{A}^{(i)} \right] - \beta \cdot \mathbb{D}_{KL}[\pi_{\theta_{new}} || \pi_{ref}] \right\} \tag{2}$$

where $\rho^{(i)} = \pi_{\theta_{new}}(o^{(i)} \mid q) / \pi_{\theta_{old}}(o^{(i)} \mid q)$, $Q$ denotes the candidate question set, and $\mathbb{D}_{KL}$ denotes the KL regularization. $\pi_{ref}$ is typically a frozen pre-trained MLLM. GRPO effectively integrates consistent policy updates and strong reward signals in a balanced way. To our knowledge, we are the first to apply GRPO to image quality understanding tasks, enabling our model to achieve robust reasoning and generalization performance without heavy reliance on extensive annotated data.

### 3.2 Overview of Q-Insight

The overall framework of Q-Insight is illustrated in Fig. 2. During training, we jointly optimize two tasks: score regression and degradation perception. Specifically, the multi-modal input for each task comprises an image paired with a task-specific question. Given these inputs, the policy model $\pi_\theta$ generates groups of answers, each accompanied by explicit reasoning steps. Subsequently, each answer is evaluated using its corresponding reward function: $R_{scr}$ for score regression, and $R_{deg}$ and $R_{lev}$ for degradation perception. After computing rewards for each group of answers, the policy model is optimized jointly via the multi-task GRPO algorithm. Additionally, a KL-divergence loss is applied to constrain deviations between the policy model $\pi_\theta$ and the reference model $\pi_{ref}$. During inference, the trained Q-Insight generates coherent reasoning processes and outputs precise answers. Further details regarding multi-task GRPO and data construction are provided in Sec. 3.3 and Sec. 3.4.

### 3.3 Multi-Task Group Relative Policy Optimization

As depicted in Fig. 2, for each input data pair, the policy model $\pi_\theta$ generates a group of $N$ responses, denoted as $\{o^{(i)}\}_{i=1}^N$. We then evaluate each of these responses using the proposed reward functions

($R_{\mathrm{scr}}$, $R_{\mathrm{deg}}$, and $R_{\mathrm{lev}}$) and obtain the overall rewards $\{r^{(i)}\}_{i=1}^{N}$. Proper reward design is crucial, as informative and carefully constructed rewards directly facilitate Q-Insight's ability to effectively learn reasoning and perception patterns, thus ensuring robust performance across multiple tasks. Specifically, Q-Insight employs a general format reward function shared across all tasks, as well as task-specific reward functions tailored to the unique characteristics of each individual task.

**Format reward** evaluates whether the reasoning steps are properly enclosed within "<think>" and "</think>" tags, and the final answer is correctly enclosed within "<answer>" and "</answer>" tags [15]. Additionally, we require that the content inside "<answer>" tags follow a JSON-like format: beginning with "{", ending with "}", and containing no additional "{" or "}" characters internally. This ensures that Q-Insight can consistently parse results across different tasks. The reward score $r_{\mathrm{fmt}}^{(i)}$ is set to 1 if the $i$-th response fulfills all the above conditions; otherwise, its reward is 0.

**Rewards for score regression task.** A standard way to quantify image quality is by using the Mean Opinion Score (MOS). Instead of directly fitting the MLLM predictions to MOS, we use MOS as a general guideline to motivate the model towards deeper reasoning and generating insightful perspectives during the process of evaluating image quality. Inspired by the treatment of mathematical reasoning tasks in DeepSeek-R1 [15], we consider the continuous MOS prediction as either correct or incorrect, thereby adopting a binary reward to avoid extremely large or small reward values. Denoting the predicted score of the $i$-th response as $\mathrm{scr}_{\mathrm{pred}}^{(i)}$ and the ground-truth score as $\mathrm{scr}_{\mathrm{gt}}$, we design the verifiable reward for scoring as follows. The reward value $r_{\mathrm{scr}}^{(i)}$ for the $i$-th response is determined by:

$$r_{\mathrm{scr}}^{(i)} = 1 \quad \text{if } |\mathrm{scr}_{\mathrm{pred}}^{(i)} - \mathrm{scr}_{\mathrm{gt}}| < \epsilon, \text{ otherwise } 0, \tag{3}$$

where $\epsilon$ is a predefined threshold. In particular, if $\epsilon$ is set to 0, the reward simplifies to exact-answer matching. Otherwise, the threshold $\epsilon$ allows the model's predicted scores to fluctuate within an acceptable range, rather than strictly requiring exact accuracy. As depicted in Fig. 2, the predicted score receives a reward of 1 if it lies within the threshold $\epsilon$ of the ground-truth MOS, and 0 otherwise.

**Rewards for degradation perception task.** We find that training solely with score labels leads to poor perception of detailed image degradations (e.g., JPEG compression). This may be because generic multimodal models are pre-trained primarily to capture high-level semantic information, causing them to ignore subtle low-level distortions. To address this issue, we jointly train the model with a degradation perception task, leveraging easily obtainable degradation labels, thus enhancing the model's sensitivity to these image degradations. In this task, the model is required to predict both the distortion class and the corresponding distortion level. Since distortion class and level are inherently discrete variables, we similarly design binary rewards for this task. Denoting the predicted distortion class and level of the $i$-th response as $\mathrm{deg}_{\mathrm{pred}}^{(i)}$ and $\mathrm{lev}_{\mathrm{pred}}^{(i)}$ respectively, we define the degradation classification reward as follows. The reward value $r_{\mathrm{deg}}^{(i)}$ is determined by:

$$r_{\mathrm{deg}}^{(i)} = 1 \quad \text{if } \mathrm{deg}_{\mathrm{pred}}^{(i)} = \mathrm{deg}_{\mathrm{gt}}, \text{ otherwise } 0. \tag{4}$$

This means the predicted distortion class receives a reward of 1 if correct, and 0 otherwise, as illustrated in Fig. 2. Similarly, the intensity perception reward $r_{\mathrm{lev}}^{(i)}$ is determined by:

$$r_{\mathrm{lev}}^{(i)} = 1 \quad \text{if } \mathrm{deg}_{\mathrm{pred}}^{(i)} = \mathrm{deg}_{\mathrm{gt}} \text{ and } \mathrm{lev}_{\mathrm{pred}}^{(i)} = \mathrm{lev}_{\mathrm{gt}}, \text{ otherwise } 0. \tag{5}$$

As depicted in Fig. 2, the predicted distortion level earns a reward of 1 only when both the predicted class and the level exactly match the ground truth; otherwise, it receives 0.

**Overall multi-task reward.** Finally, the overall reward of $i$-th response is calculated as:

$$r^{(i)} = r_{\mathrm{fmt}}^{(i)} + \mathbb{1}_{\mathrm{scr}} \cdot r_{\mathrm{scr}}^{(i)} + \mathbb{1}_{\mathrm{deg}} \cdot \left( \alpha_1 \cdot r_{\mathrm{deg}}^{(i)} + \alpha_2 \cdot r_{\mathrm{lev}}^{(i)} \right), \tag{6}$$

where $\mathbb{1}_{\mathrm{scr}}$ equals 1 if the score regression task is selected (and 0 otherwise), and similarly, $\mathbb{1}_{\mathrm{deg}}$ equals 1 if the degradation perception task is selected (and 0 otherwise). Note that the reasoning process illustrated in Fig. 2 emerges naturally from the model's internal capability, without relying on external constraints or additional annotated data. After computing rewards for all generated responses $\{r^{(1)}, r^{(2)}, ..., r^{(N)}\}$, the policy model is updated following Eqs. (1) and (2). With this flexible design, Q-Insight can seamlessly switch between tasks and jointly optimize them during

Table 1: **PLCC / SRCC comparison on the score regression tasks** between our Q-Insight and other competitive IQA methods. All methods except handcrafted ones are trained on the KonIQ dataset. The best and second-best results of each test setting are highlighted in **bold red** and underlined blue.

| Category | Methods | KonIQ | SPAQ | KADID | PIPAL | LiveW | AGIQA | CSIQ | AVG. |
|---|---|---|---|---|---|---|---|---|---|
| Handcrafted | NIQE [29] (SPL 2012) | 0.533 /0.530 | 0.679 /0.664 | 0.468 /0.405 | 0.195 /0.161 | 0.493 /0.449 | 0.560 /0.533 | 0.718 /0.628 | 0.521 /0.481 |
| | BRISQUE [28] (TIP 2012) | 0.225 /0.226 | 0.490 /0.406 | 0.429 /0.356 | 0.267 /0.232 | 0.361 /0.313 | 0.541 /0.497 | 0.740 /0.556 | 0.436 /0.369 |
| Non-MLLM Deep-learning | NIMA [47] (TIP 2018) | 0.896 /0.859 | 0.838 /0.856 | 0.532 /0.535 | 0.390 /0.399 | 0.814 /0.771 | 0.715 /0.654 | 0.695 /0.649 | 0.697 /0.675 |
| | HyperIQA [44] (CVPR 2020) | 0.917 /0.906 | 0.791 /0.788 | 0.506 /0.468 | 0.410 /0.403 | 0.772 /0.749 | 0.702 /0.640 | 0.752 /0.717 | 0.693 /0.667 |
| | DBCNN [67] (TCSVT 2020) | 0.884 /0.875 | 0.812 /0.806 | 0.497 /0.484 | 0.384 /0.381 | 0.773 /0.755 | 0.730 /0.641 | 0.586 /0.572 | 0.667 /0.645 |
| | MUSIQ [19] (ICCV 2021) | 0.924 /0.929 | 0.868 /0.863 | 0.575 /0.556 | 0.431 /0.431 | 0.789 /0.830 | 0.722 /0.630 | 0.771 /0.710 | 0.726 /0.707 |
| | CLIP-IQA+ [49] (AAAI 2023) | 0.909 /0.895 | 0.866 /0.864 | 0.653 /0.654 | 0.427 /0.419 | 0.832 /0.805 | 0.736 /0.685 | 0.772 /0.719 | 0.742 /0.720 |
| | ManIQA [56] (CVPR 2022) | 0.849 /0.834 | 0.768 /0.758 | 0.499 /0.465 | 0.457 /0.452 | 0.849 /0.832 | 0.723 /0.636 | 0.623 /0.627 | 0.681 /0.658 |
| MLLM-based | C2Score [75] (NeurIPS 2024) | 0.923 /0.910 | 0.867 /0.860 | 0.500 /0.453 | 0.354 /0.342 | 0.786 /0.772 | 0.777 /0.671 | 0.735 /0.705 | 0.706 /0.673 |
| | Qwen-SFT [1] (Arxiv 2025) | 0.889 /0.866 | 0.874 /0.875 | 0.668 /0.663 | 0.473 /0.442 | 0.734 /0.728 | **0.813** /0.739 | 0.674 /0.650 | 0.732 /0.709 |
| | Q-Align [53] (ICML 2024) | 0.941 /0.940 | 0.886 /0.887 | 0.674 /0.684 | 0.403 /0.419 | 0.853 /0.860 | 0.772 /0.735 | 0.671 /0.737 | 0.705 /0.752 |
| | DeQA [58] (CVPR 2025) | **0.953** /**0.941** | 0.895 /0.896 | 0.694 /0.687 | 0.472 /**0.478** | 0.892 /**0.879** | 0.809 /0.729 | 0.787 /0.744 | 0.786 /0.765 |
| | **Q-Insight** (Ours) | 0.933 /0.916 | **0.907** /**0.905** | **0.742** /**0.736** | **0.486** /0.474 | **0.893** /0.865 | 0.811 /**0.764** | **0.870** /**0.824** | **0.806** /**0.783** |

training. During inference, the trained policy model can directly perform image quality understanding without requiring additional fine-tuning. Experimental results presented in Tabs. 3 and 4 and Fig. 4 further demonstrate that jointly addressing the score regression and degradation perception tasks substantially improves performance, highlighting the beneficial interactions between these two tasks.

### 3.4 Data Construction

We construct multi-modal training data to jointly train Q-Insight on the score regression and degradation perception tasks. The prompts designed for each task are detailed in Tab. A in the appendix. For the score regression task, the input includes a task-specific prompt and the image to be rated, with the Mean Opinion Score (MOS) serving as the guideline to calculate the corresponding reward. In the degradation perception task, the input consists of a prompt and an image characterized by a specific distortion class and severity level. There are five distortion categories: "noise", "blur", "JPEG", "darken", and "null", where "null" indicates no distortion. Each distortion type has five severity levels: "slight", "moderate", "obvious", "serious", and "catastrophic". The distortion class and corresponding severity level constitute the ground-truth labels to calculate the degradation classification and intensity perception rewards. Overall, our carefully-designed GRPO-based framework and multi-task training strategy help Q-Insight achieve robust reasoning and perception capability even from limited annotated labels. More importantly, this flexible approach facilitates effective generalization to various low-level vision applications, as demonstrated by extensive experiments in Sec. 4, clearly highlighting the advantages of our framework in addressing practical vision tasks.

## 4 Experiments

### 4.1 Experimental Setup

**Datasets and Metrics.** For the score regression task, we use diverse IQA datasets across four categories: (a) in-the-wild datasets, including KonIQ [16], SPAQ [11], and LIVE-Wild [12]; (b)

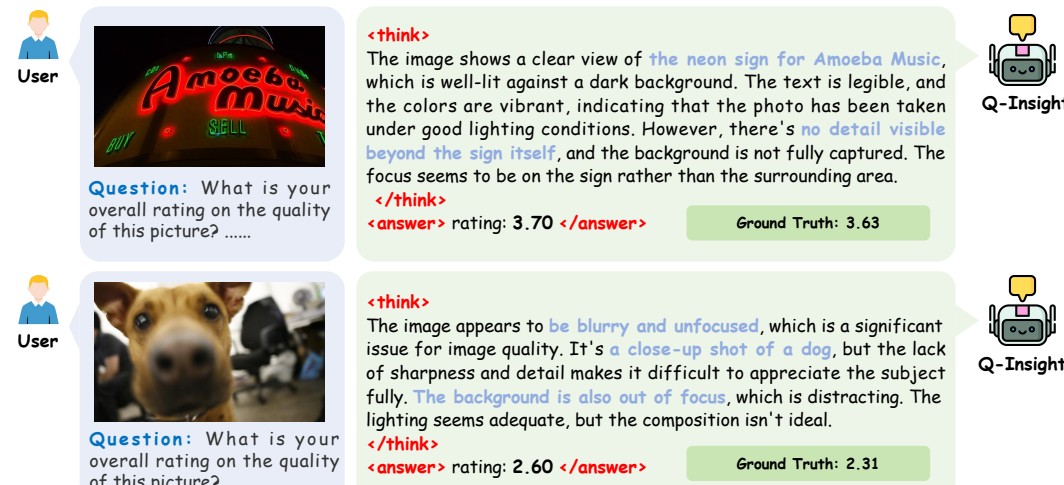

Figure 3: **Score rating and explanation results of our Q-Insight.** Q-Insight is capable of recognizing text, analyzing the lighting and shading conditions of an image, and understanding its composition.

Table 2: **Distortion prediction accuracy (Deg. Acc.) and degradation level accuracy (Lev. Acc.)** comparison between our Q-Insight and AgenticIR [76]. Our method outperforms AgenticIR across all degradations, especially in Noise and JPEG Compression.

| Method | Metrics | Noise | Blur | JPEG | Darken | Null | Average |
|---|---|---|---|---|---|---|---|
| AgenticIR [76] (ICLR 2025) | Deg. Acc. | 0.4646 | 0.8390 | 0.0135 | 0.7478 | **0.9339** | 0.5998 |
| | Lev. Acc. | 0.1858 | 0.3219 | 0.0000 | 0.2611 | - | 0.1922 |
| Q-Insight (Ours) | Deg. Acc. | **1.0000** | **0.9756** | **1.0000** | **0.9027** | 0.7603 | **0.9277** |
| | Lev. Acc. | **0.5973** | **0.4438** | **0.5541** | **0.3230** | - | **0.4796** |

synthetic distortion datasets, including KADID [22] and CSIQ [20]; (c) model-processed distortions, including PIPAL [14]; and (d) AI-generated images from AGIQA [21]. Following [58], we split KonIQ into training and test sets, with approximately 7000 training images. Mean Opinion Scores (MOS) across these datasets are normalized into the range $[1, 5]$. The remaining datasets are exclusively used to evaluate the model's out-of-distribution (OOD) generalization capability. For degradation perception task, we randomly select 7000 images from DQ-495K [59] that contain a single distortion for training, with an additional 1000 images reserved for testing. We adopt the pearson linear correlation coefficient (PLCC) and spearman rank-order correlation coefficient (SRCC) as metrics to evaluate performance on score regression task, following [19, 53, 58]. For degradation perception, we use the accuracy of distortion class and degradation level as evaluation metrics.

**Implementation Details.** We adopt Qwen-2.5-VL-7B-Instruct [1] as our base model. In the GRPO algorithm, the generation number $N$ is set to 8, the weight of KL divergence penalty $\beta$ is set to $1 \times 10^{-3}$, while the weights $\alpha_1$ and $\alpha_2$ are set to 0.25 and 0.75, respectively. The threshold $\epsilon$ is set to 0.35. We employ AdamW [26] as the optimizer, using an initial learning rate of $1 \times 10^{-6}$ that linearly decays to $1 \times 10^{-9}$ during training. The model is trained for 10 epochs with a total batch size of 128. Training is completed in approximately one day using 16 NVIDIA A100 GPUs.

## 4.2 Score Regression

We first evaluate our Q-Insight on the score regression task. We compare Q-Insight with handcrafted methods NIQE [29] and BRISQUE [28]; non-MLLM deep-learning methods including NIMA [47], MUSIQ [19], CLIP-IQA+ [49], and ManIQA [56]; and recent MLLM-based methods such as C2Score [75], Q-Align [53], DeQA-Score [58], and a supervised fine-tuned Qwen [1]. For a fair comparison, all methods (except handcrafted ones) are trained on the KonIQ dataset, and all MLLM-based methods utilize approximately 7B parameters. The comparison results in terms of PLCC and SRCC between Q-Insight and other IQA methods are presented in Tab. 1. Compared with the state-of-the-art method DeQA-Score, our Q-Insight performs slightly worse on the in-domain KonIQ

Table 3: **Ablation study on the score regression task** between multi-task and single-task training. Q-Insight with joint-training significantly outperforms score-only training on PLCC / SRCC metrics.

| Method | KonIQ | SPAQ | KADID | PIPAL | LiveW | AGIQA | CSIQ | AVG. |
|---|---|---|---|---|---|---|---|---|
| Ours (Score-Only) | 0.918 /0.895 | 0.903 /0.899 | 0.702 /0.702 | 0.458 /0.435 | 0.870 /0.839 | **0.816** /**0.766** | 0.685 /0.640 | 0.765 /0.739 |
| Ours (Joint-Training) | **0.933** /**0.916** | **0.907** /**0.905** | **0.742** /**0.736** | **0.486** /**0.474** | **0.893** /**0.865** | 0.811 /0.764 | **0.870** /**0.824** | **0.806** /**0.783** |

Table 4: **Ablation study on the degradation perception task** between multi-task and single-task training. Jointly training with score regression improves the accuracy of degradation perception.

| Method | Metrics | Noise | Blur | JPEG | Darken | Null | Average |
|---|---|---|---|---|---|---|---|
| Ours (Degradation-Only) | Deg. Acc. | 0.9867 | 0.9268 | 0.9685 | 0.8805 | 0.5702 | 0.8960 |
| | Lev. Acc. | 0.4343 | 0.3951 | 0.3108 | 0.2567 | - | 0.3492 |
| Ours (Joint-Training) | Deg. Acc. | **1.0000** | **0.9756** | **1.0000** | **0.9027** | **0.7603** | **0.9277** |
| | Lev. Acc. | **0.5973** | **0.4438** | **0.5541** | **0.3230** | - | **0.4796** |

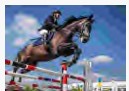

**Question:** What's your overall rating on the quality of this picture? The rating should be a float between 1 and 5, rounded to two decimal places, with 1 representing very poor quality and 5 representing excellent quality.

☐ **w/o Joint Training**

`<think>`
The image has good lighting and vibrant colors, which contribute to its overall appeal. However, there is noticeable noise present, which can detract from the image quality. The horse and rider appear slightly blurry, indicating motion or a lower resolution. Despite these issues, the image still maintains a certain level of clarity and detail, especially in the background. Therefore, I would rate the image quality as moderate.
`</think>`
`<answer>` rating: **3.50** `</answer>`

☐ **Joint Training**

`<think>`
The image appears to be a digital illustration rather than a photograph, which can affect its quality. It has a somewhat pixelated appearance, suggesting it might not be high resolution. The colors are vibrant but the details are not sharp. The composition is interesting, capturing an action scene well, but the lack of detail and clarity brings down the overall quality.
`</think>`
`<answer>` rating: **2.70** `</answer>`

Figure 4: **Subjective ablation comparison** between joint multi-task training and w/o joint training on the explanation of image scoring. With joint training, our method can better perceive degradation cues in images (such as pixelated appearance), thereby improving the accuracy of quality assessment.

dataset. However, **on out-of-distribution (OOD) datasets, Q-Insight consistently outperforms all baseline methods across nearly all benchmarks**, achieving approximately 0.02 improvements in both PLCC and SRCC. This demonstrates the effectiveness and strong generalization capability of our approach. Fig. 3 illustrates two cases showing the reasoning capability in the score regression task. Specifically, our method goes beyond merely outputting numerical scores and provides detailed, structured reasoning. In the first case (top of Fig. 3), Q-Insight correctly identifies and analyzes textual information displayed on a neon sign, thoroughly examining details such as lighting conditions. In the second case (bottom of Fig. 3), Q-Insight demonstrates its strength in interpreting image composition aspects, such as the arrangement of visual elements and the primary focal point of the image. These examples further illustrate how Q-Insight advances beyond score regression task, offering valuable insights into image quality by examining various perceptual factors from multiple perspectives, ultimately contributing to a comprehensive understanding of image quality.

## 4.3 Distortion Perception

We further evaluate Q-Insight on the distortion perception task, comparing it with AgenticIR [76], which fine-tunes an MLLM to perform a similar distortion prediction function. The comparative results are presented in Tab. 2. Notably, AgenticIR requires sequential queries for each possible distortion type, whereas Q-Insight identifies distortion types **using only a single query**. Q-Insight consistently outperforms AgenticIR across nearly all distortion categories, resulting in significantly higher average accuracy. However, for the "null" category (no distortion), our performance is slightly

Table 5: **Ablation study on the threshold $\epsilon$ for the score regression task.** Q-Insight demonstrates robust and stable performance, indicating it does not require careful tuning of the threshold $\epsilon$.

| Method | KonIQ | SPAQ | KADID | PIPAL | LiveW | AGIQA | CSIQ |
|---|---|---|---|---|---|---|---|
| $\epsilon = 0.15$ | 0.918/0.900 | 0.903/0.903 | 0.749/0.744 | 0.443/0.441 | 0.876/0.847 | **0.835**/0.764 | 0.857/0.793 |
| $\epsilon = 0.25$ | 0.929/0.911 | 0.902/0.902 | **0.750/0.752** | 0.472/0.471 | 0.884/0.855 | 0.823/0.765 | 0.889/0.844 |
| $\epsilon = 0.30$ | 0.932/0.912 | 0.906/0.903 | 0.742/0.738 | 0.471/0.460 | 0.887/0.856 | 0.828/**0.773** | 0.868/0.822 |
| $\epsilon = 0.40$ | 0.930/0.912 | 0.904/0.901 | 0.721/0.723 | 0.446/0.428 | 0.880/0.853 | 0.819/0.769 | 0.873/0.831 |
| $\epsilon = 0.50$ | 0.928/0.906 | **0.906**/0.902 | **0.750**/0.746 | 0.475/0.472 | 0.885/0.854 | 0.821/0.768 | **0.894/0.850** |
| $\epsilon = 0.35$ **(Ours)** | **0.933/0.916** | **0.907/0.905** | 0.742/0.736 | **0.486/0.474** | **0.893/0.865** | 0.811/0.764 | 0.870/0.824 |

Figure 5: **Image comparison reasoning results** of our Q-Insight and DepictQA [60]. Q-Insight outperforms DepictQA in comprehensive content understanding and accurate degradation perception.

lower than AgenticIR, possibly because the reasoning mechanism of Q-Insight is more sensitive to the presence of potential distortions. Overall, these results highlight Q-Insight's ability to efficiently and accurately perceive image distortions through a concise and unified reasoning framework.

## 4.4 Ablation Studies

**Effect of multi-task training.** To validate the effectiveness of multi-task training, we compare our jointly trained Q-Insight model with two single-task variants, each trained independently on a single task. The comparison results are presented in Tabs. 3 and 4. As shown in Tab. 3, the jointly trained Q-Insight significantly outperforms the score-only variant on nearly all datasets, especially on datasets involving synthetic distortions (KADID [22], CSIQ [20]) and those containing model-generated distortions (PIPAL [14]). This demonstrates that incorporating the degradation perception task can effectively enhance performance in the score regression task. Fig. 4 further presents the benefits of multi-task training, showing that Q-Insight can precisely identify detailed degradations such as pixel-level artifacts, thereby improving overall accuracy in quality assessment. Similarly, Tab. 4 indicates that in the degradation perception task, our jointly trained model consistently surpasses the degradation-only variant across all distortion types. This suggests that the score regression task also positively contributes to degradation perception capabilities. These experimental results verify the mutual benefit and effectiveness of the proposed multi-task training strategy. Moreover, our findings clearly show that the visual quality understanding potential of MLLMs can be significantly improved through carefully designed training tasks and learning objectives.

**Ablation on the score threshold $\epsilon$.** Introducing the threshold allows the model's predictions to vary within an acceptable margin. Tab. 5 reports the ablation results for different choices of $\epsilon$. Q-Insight consistently achieves robust and stable performance across various threshold values, demonstrating that its effectiveness does not depend on careful tuning of $\epsilon$.

Table 6: **Accuracy and PLCC / SRCC results of the reference-based comparison task on the SRbench [6].** Reg-Acc and Gen-Acc represent the accuracy between regression-based and generation-based restoration methods, respectively. Q-Insight outperforms score- and description-based methods.

| Category | Method | Reg-Acc | Gen-Acc | Overall-Acc | PLCC | SRCC |
|---|---|---|---|---|---|---|
| Score-Based | PSNR | 80.07% | 41.70% | 34.70% | – | – |
| | SSIM [50] (TIP 04) | 83.00% | 45.30% | 37.40% | – | – |
| | LPIPS [66] (CVPR 18) | 82.00% | 63.90% | 65.80% | – | – |
| | A-FINE [6] (CVPR 25) | 83.30% | **78.90%** | 82.40% | – | – |
| Description-Based | DepictQA [60] (ECCV 24) | 73.00% | 61.64% | 62.96% | 0.3457 | 0.3412 |
| | Q-Insight (Zero-Shot) | 78.67% | 68.64% | 75.51% | 0.6385 | 0.6297 |
| | Q-Insight (Trained) | **85.67%** | 77.78% | **82.80%** | 0.7627 | 0.7614 |

## 4.5 Image Comparison Reasoning

Our Q-Insight effectively generalizes to zero-shot image comparison reasoning tasks in both reference-based and non-reference-based scenarios, as illustrated in Figs. 1 and 5. Specifically, Fig. 1 shows a reference-based comparison scenario, where the reference image is of lower quality, and Images A and B are outputs generated by two different super-resolution methods. Fig. 5 demonstrates Q-Insight's superiority over DepictQA [60], highlighting its enhanced content understanding and precise perception of degradations. These examples illustrate Q-Insight's robust generalization ability, largely enabled by its RL-based framework and multi-task training strategy.

Furthermore, the comparison reasoning performance can be further boosted by training on a small number of labeled comparison pairs using reinforcement learning. Specifically, we randomly sample 5k data pairs from the DiffIQA [6] dataset, where each pair is labeled only with comparison results, without any textual descriptions. Results are shown in Tab. 6. Our Q-Insight consistently surpasses all score-based and description-based methods in terms of overall accuracy and PLCC/SRCC metrics, demonstrating its promising applicability to various image enhancement tasks. Notably, even the zero-shot version of Q-Insight substantially outperforms DepictQA [60], despite the latter relying on large-scale textual datasets. Additionally, A-Fine [6] utilizes more than 200k data pairs collected from four different datasets, combined with a complex three-stage training pipeline, thus requiring over 40 times more data and considerable effort to develop optimal training strategies. In contrast, Q-Insight achieves superior performance through relatively straightforward yet highly effective visual reinforcement learning. Further details and additional results are provided in Sec. B of the Appendix.

## 5   Conclusion

In this paper, we introduce Q-Insight, a novel GRPO-based model for comprehensive image quality understanding. It jointly optimizes score regression and degradation perception tasks using only a limited amount of labeled data. Unlike traditional methods that rely on extensive textual annotations or purely numerical scoring, our framework combines numerical accuracy with interpretative reasoning, significantly improving the perceptual analysis capabilities of image quality models. Extensive experiments show that Q-Insight consistently outperforms existing state-of-the-art methods across various datasets and tasks, demonstrating impressive zero-shot generalization and superior comparison reasoning capability. Looking ahead, Q-Insight can extend its capabilities to a wide range of tasks, such as image aesthetic assessments, and serve as a powerful discriminative signal to improve image enhancement models. As a unified model for scoring, perception, comparison, and reasoning, Q-Insight can act as a central hub, coordinating image reconstruction tools and providing valuable insights into the enhancement process. This integrated and automated system has the potential to revolutionize image quality understanding and enhancement, providing a unified solution that can transform how image quality is evaluated, improved, and applied across various fields.

**Limitations.** While achieving promising performance, Q-Insight focuses primarily on natural images. Extending to AI-generated images and videos remains essential and is reserved for future exploration. Besides, using a fixed threshold and discrete distortion levels is not the most elegant solution and may allow for more principled approaches. These issues warrant further exploration.

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

# A Designed Prompts

The prompts designed for each task are detailed in Tab. A. Specifically, a general system prompt is shared across tasks, which encourages the model to explicitly output its reasoning process and provide structured responses. This general prompt is supplemented by task-specific prompts tailored for score regression and degradation perception, respectively. For the score regression task, the input includes a task-specific prompt and the image to be rated, with the Mean Opinion Score (MOS) serving as the ground-truth. In the degradation perception task, the input consists of a prompt and an image characterized by a specific distortion class and severity level. We define five distortion categories: "noise", "blur", "JPEG", "darken", and "null", where "null" indicates no distortion. Each distortion type has five severity levels: "slight", "moderate", "obvious", "serious", and "catastrophic". The distortion class and corresponding severity level constitute the ground-truth labels. Additionally, for the comparative reasoning scenario, the inputs include a prompt, two images to be compared, and an optional reference image.

Table A: **Prompts for Different Tasks.** The system prompt is shared across all tasks, while task-specific prompts are additionally designed for each individual task.

---

**System Prompt:** A conversation between User and Assistant. The user asks a question, and the Assistant solves it. The assistant first thinks about the reasoning process in the mind and then provides the user with the answer. The reasoning process and answer are enclosed within <think> </think> and <answer> </answer> tags, respectively, i.e., <think> reasoning process here </think><answer> answer here </answer>.

---

**Prompt for Score Regression Task:** What is your overall rating on the quality of this picture? The rating should be a float between 1 and 5, rounded to two decimal places, with 1 representing very poor quality and 5 representing excellent quality. Return the final answer in JSON format with the following keys: "rating": The score.

---

**Prompt for Degradation Perception Task:** Analyze the given image and determine if it contains any of the following distortions: "noise", "compression", "blur", or "darken". If a distortion is present, classify its severity as "slight", "moderate", "obvious", "serious", or "catastrophic". Return the result in JSON format with the following keys: "distortion_class": The detected distortion (or "null" if none). and "severity": The severity level (or "null" if none).

---

**Prompt for Non-Reference-Based Image Comparison:** Given Image A: <image_A> and <Image_B>, assess the visual quality of both the Image A and the Image B, explain and justify which one is better considering composition and degradation. Your answer should be "Image A" or "Image B".

---

**Prompt for Reference-Based Comparison Reasoning:** Given a low-quality reference image and two enhanced outputs. Reference Image: <ref_image>, Image A: <image_A> and Image B: <image_B>. Decide which enhanced image is superior or if they are comparable. Evaluate based on: 1) fidelity and consistency with the reference image; 2) overall perceptual quality. Return exactly one of: "Image A", "Image B", or "Similar".

---

# B Experimental Setups and More Results of Comparison Reasoning

## B.1 Datasets

For training, we use the DiffIQA [6] dataset, which contains approximately 180k reference–test image pairs generated by applying diffusion-based enhancement methods to reference images of varying quality. Each reference image is paired with multiple test images, and human annotators provide preference labels through triplet-based comparisons, resulting in roughly 180k comparison pairs. Notably, A-Fine [6] aggregates datasets from DiffIQA, TID2013 [35], KADID [22], and PIPAL [14], totaling over 200k comparison pairs for training. In contrast, we randomly sample only 5k pairs from DiffIQA for our training.

For evaluation, we adopt SRIQA-Bench [6], a benchmark specifically designed to evaluate the generalization ability of IQA models in real-world super-resolution (SR) scenarios. This dataset includes 100 low-resolution (LR) reference images, each enhanced by 10 distinct SR methods encompassing both regression-based and generative models. Human raters perform exhaustive pairwise comparisons within each image group, with each comparison annotated by at least 10 annotators to ensure reliability. Since no ground-truth high-resolution reference images are provided, models must evaluate perceptual quality based solely on the degraded LR images. Thus, SRIQA-Bench poses a challenging scenario to rigorously assess the robustness of full-reference IQA models under imperfect reference conditions. we report Reg-Acc, Gen acc, and overall ACC, which denote pairwise ranking accuracy on regression-based SR methods, generation-based SR methods, and all SR outputs, respectively. These metrics measure alignment between model predictions and human judgments under varying super-resolution styles and distortion characteristics. We also report the PLCC and SRCC of Description-based methods.

## B.2 Reward Design and Training Details

In the image comparison task, the model is expected to determine which image is superior or if they are comparable. Denote the predicted score of the $i$-th response as $\text{res}_{\text{pred}}^{(i)}$, we design a comparison reward function $R_{\text{comp}}$ as follows. The reward value $r_{\text{comp}}^{(i)}$ for the $i$-th response is determined by :

$$r_{\text{comp}}^{(i)} = 1 \quad \text{if } \text{res}_{\text{pred}}^{(i)} = \text{res}_{\text{comp}}, \text{ otherwise } 0. \tag{7}$$

Finally, the overall reward of $i$-th response is calculated as:

$$r^{(i)} = r_{\text{fmt}}^{(i)} + r_{\text{comp}}^{(i)}, \tag{8}$$

where $r_{\text{fmt}}^{(i)}$ is the format reward. For training, we employ AdamW [26] as the optimizer, using an initial learning rate of $1 \times 10^{-6}$ that linearly decays to $1 \times 10^{-9}$ during training. The model is trained for 8 epochs with a total batch size of 32. Training is completed in approximately 20 hours using 16 NVIDIA A100 GPUs.

## B.3 More Qualitative Results

We provide quantitative results in Tab. 6 in the main paper. Here we present additional qualitative results of reference-based image comparison reasoning in Figs. A, B, C, and D. It can be observed that, despite training on only 5k comparison pairs, Q-Insight surpasses DepictQA [60] by more effectively recognizing subtle image details and distortions, thereby producing more accurate outcomes.

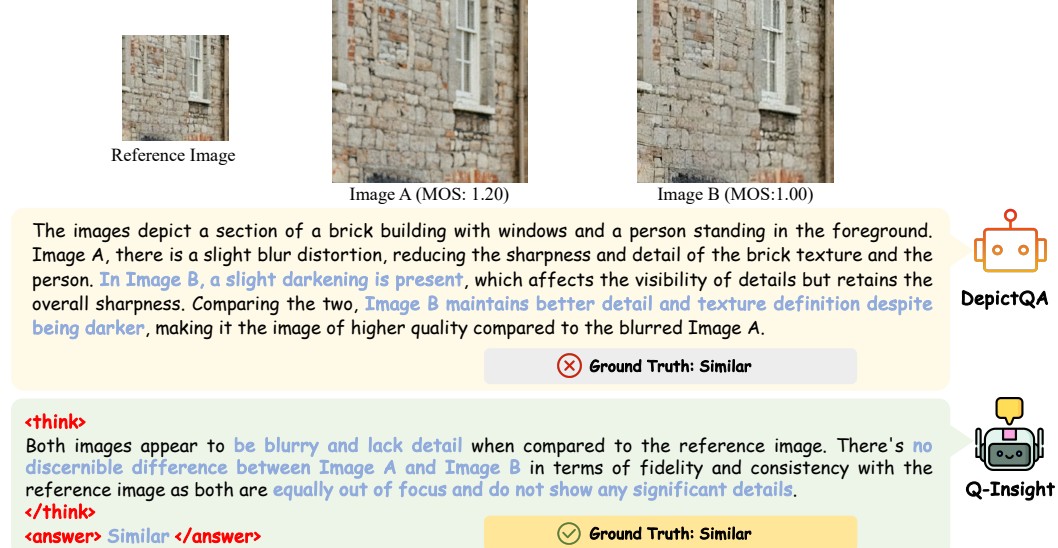

Reference Image

Image A (MOS: 1.20)    Image B (MOS:1.00)

The images depict a section of a brick building with windows and a person standing in the foreground. Image A, there is a slight blur distortion, reducing the sharpness and detail of the brick texture and the person. In Image B, a slight darkening is present, which affects the visibility of details but retains the overall sharpness. Comparing the two, Image B maintains better detail and texture definition despite being darker, making it the image of higher quality compared to the blurred Image A.

DepictQA

⊗ Ground Truth: Similar

<think>
Both images appear to be blurry and lack detail when compared to the reference image. There's no discernible difference between Image A and Image B in terms of fidelity and consistency with the reference image as both are equally out of focus and do not show any significant details.
</think>
<answer> Similar </answer>

Q-Insight

⊘ Ground Truth: Similar

Figure A: **Reference-based image comparison reasoning results** of our Q-Insight and Depic-tQA [60] on image "online4" from SRBench [6].

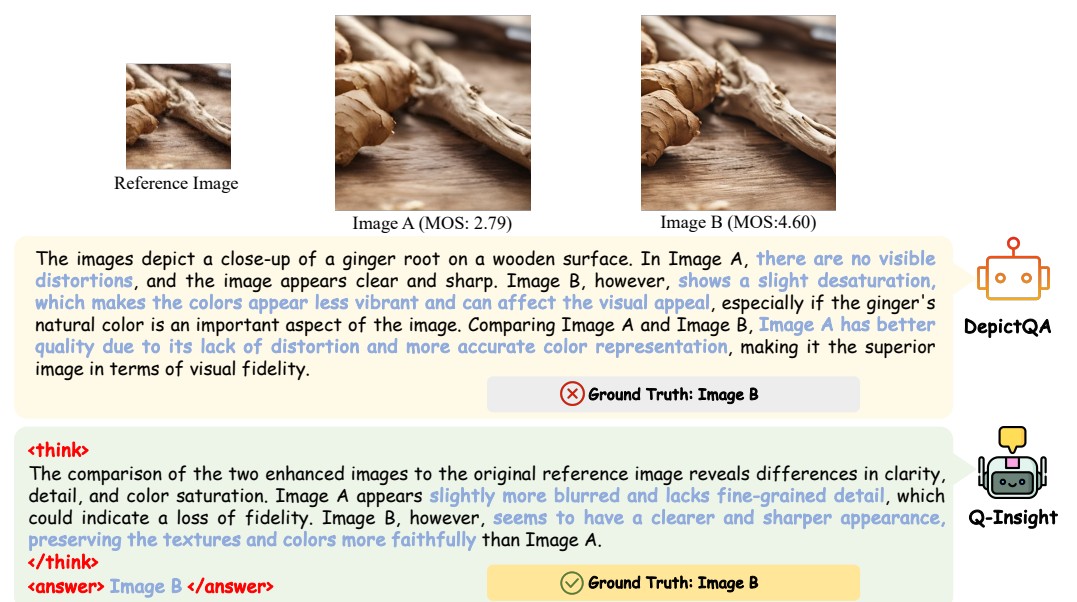

Reference Image

Image A (MOS: 2.79)    Image B (MOS:4.60)

The images depict a close-up of a ginger root on a wooden surface. In Image A, there are no visible distortions, and the image appears clear and sharp. Image B, however, shows a slight desaturation, which makes the colors appear less vibrant and can affect the visual appeal, especially if the ginger's natural color is an important aspect of the image. Comparing Image A and Image B, Image A has better quality due to its lack of distortion and more accurate color representation, making it the superior image in terms of visual fidelity.

DepictQA

⊗ Ground Truth: Image B

<think>
The comparison of the two enhanced images to the original reference image reveals differences in clarity, detail, and color saturation. Image A appears slightly more blurred and lacks fine-grained detail, which could indicate a loss of fidelity. Image B, however, seems to have a clearer and sharper appearance, preserving the textures and colors more faithfully than Image A.
</think>
<answer> Image B </answer>

Q-Insight

⊘ Ground Truth: Image B

Figure B: **Reference-based image comparison reasoning results** of our Q-Insight and Depic-tQA [60] on image "online23" from SRBench [6].

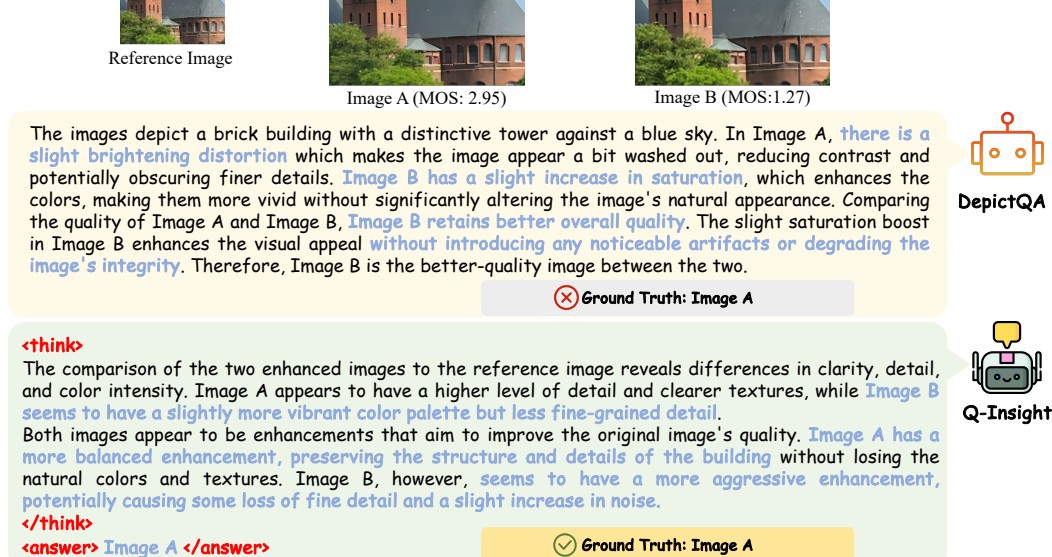

Reference Image

Image A (MOS: 2.95)    Image B (MOS:1.27)

The images depict a brick building with a distinctive tower against a blue sky. In Image A, there is a slight brightening distortion which makes the image appear a bit washed out, reducing contrast and potentially obscuring finer details. Image B has a slight increase in saturation, which enhances the colors, making them more vivid without significantly altering the image's natural appearance. Comparing the quality of Image A and Image B, Image B retains better overall quality. The slight saturation boost in Image B enhances the visual appeal without introducing any noticeable artifacts or degrading the image's integrity. Therefore, Image B is the better-quality image between the two.

DepictQA

⊗ Ground Truth: Image A

Q-Insight

**<think>**
The comparison of the two enhanced images to the reference image reveals differences in clarity, detail, and color intensity. Image A appears to have a higher level of detail and clearer textures, while Image B seems to have a slightly more vibrant color palette but less fine-grained detail.
Both images appear to be enhancements that aim to improve the original image's quality. Image A has a more balanced enhancement, preserving the structure and details of the building without losing the natural colors and textures. Image B, however, seems to have a more aggressive enhancement, potentially causing some loss of fine detail and a slight increase in noise.
**</think>**
**<answer>** Image A **</answer>**

⊘ Ground Truth: Image A

Figure C: **Reference-based image comparison reasoning results** of our Q-Insight and Depic-tQA [60] on image "realworld33" from SRBench [6].

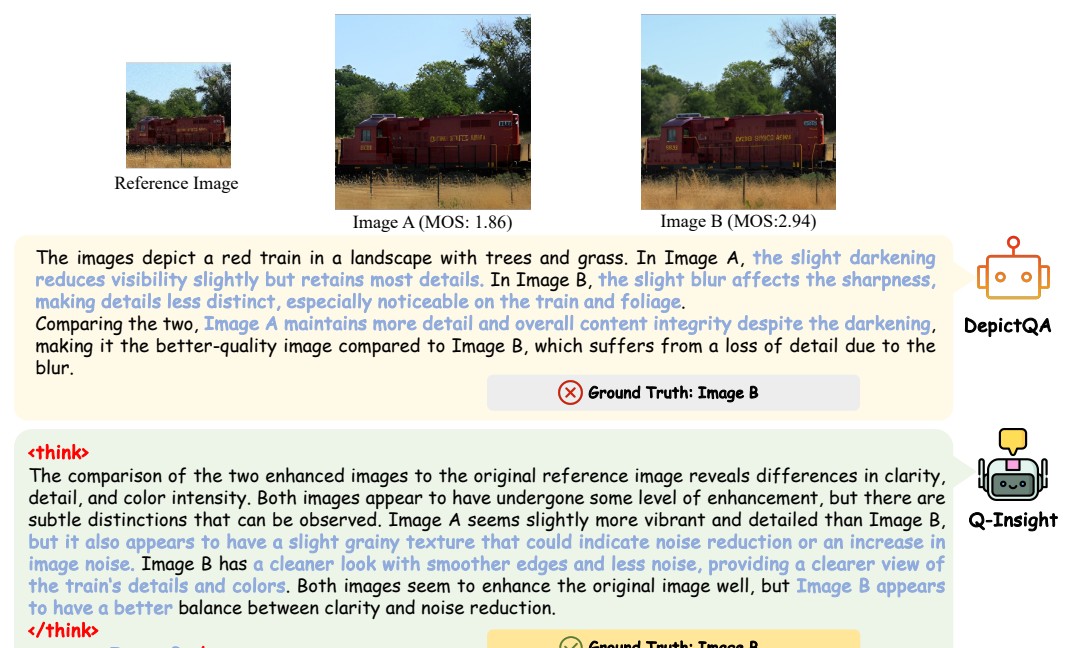

Reference Image

Image A (MOS: 1.86)    Image B (MOS:2.94)

The images depict a red train in a landscape with trees and grass. In Image A, the slight darkening reduces visibility slightly but retains most details. In Image B, the slight blur affects the sharpness, making details less distinct, especially noticeable on the train and foliage.
Comparing the two, Image A maintains more detail and overall content integrity despite the darkening, making it the better-quality image compared to Image B, which suffers from a loss of detail due to the blur.

DepictQA

⊗ Ground Truth: Image B

Q-Insight

**<think>**
The comparison of the two enhanced images to the original reference image reveals differences in clarity, detail, and color intensity. Both images appear to have undergone some level of enhancement, but there are subtle distinctions that can be observed. Image A seems slightly more vibrant and detailed than Image B, but it also appears to have a slight grainy texture that could indicate noise reduction or an increase in image noise. Image B has a cleaner look with smoother edges and less noise, providing a clearer view of the train's details and colors. Both images seem to enhance the original image well, but Image B appears to have a better balance between clarity and noise reduction.
**</think>**
**<answer>** Image B **</answer>**

⊘ Ground Truth: Image B

Figure D: **Reference-based image comparison reasoning results** of our Q-Insight and Depic-tQA [60] on image "srtest55" from SRBench [6].

