# OpenReview forum: "Q-Insight: Understanding Image Quality via Visual Reinforcement Learning"
_NeurIPS.cc/2025/Conference — NeurIPS 2025 spotlight_

### Official Review · Reviewer_51Yp · 2025-06-25

**Clarity:** 4
**Significance:** 4
**Originality:** 4
**Rating:** 5
**Confidence:** 5

**Summary:**

This work mainly addresses the two challenges of current MLLM-based image quality assessment methods: less interpretation of estimated scores and the data-hungry issue. This work applied reinforcement learning-based finetuning on MLLM for image quality understanding. Specifically, three tasks are considered: quality score regression, degradation perception (estimating distortion type and level), and comparison reasoning. The Group Relative Policy Optimization is utilized as the reinforcement learning algorithm to solve the problem. The proposed framework demonstrates a great out-of-distribution generalization performance, and the qualitative results show the proposed framework's potential for a deep understanding of image quality.

**Questions:**

Corresponding to the five listed major weaknesses, my questions or suggestions are listed as follows:

1) Could you conduct preliminary subjective experiments on comparing the reasoning capability for DepictQA and Q-Insight? If the experiment design and results are satisfactory, I will increase my evaluation score.

2) Could you analyze the possibility of extending the proposed framework to more complicated distortion types? Furthermore, if you could show the potential for this task by extra experimental results, I will increase my evaluation score.

3) Could you explain why smaller thresholds cannot lead to a better score estimation performance?

4) I suggest the authors add one extra experiment to demonstrate the stability of the RL algorithm.

5) I suggest the authors optimize the experiments design for comparing non-MLLM methods and MLLM methods.

**Ethical Concerns:**

["NO or VERY MINOR ethics concerns only"]

**Final Justification:**

The authors have addressed my concerns well, especially regarding the evaluation of the reasoning capability of the proposed framework. I also reviewed the discussion between the authors and other reviewers. It seems like the other reviewers are also satisfied with the rebuttal. Therefore, I decided to change my rating from 3 to 5.

**Limitations:**

yes

**Quality:**

4

**Strengths And Weaknesses:**

Strengths:

1) This is a high quality work, solid technical contribution, well-written and easy-to-follow paper writing, well-explained concepts, and the experimental results support the claims proposed in the paper.

2) The proposed method has been proven data efficient for achieving competitive results with few data points.

3) The experimental results show a good generalization capability of the proposed framework on quality score regression tasks. The qualitative results are carefully presented with rich details.

Major Weaknesses:

1) In the paper, the comparison reasoning capability is demonstrated by several qualitative cases comparing the reasoning processes of DepictQA and Q-Insight, which is not sufficient. A large number of well-organized subjective experiments would be needed for a thorough comparison between these two methods, which is a common technique for evaluation in the field of quality assessment.

2) The current setting on distortion types is too simple, only five distortion types are selected, and one of them is "null". From Table 4, we can see the model almost achieves 100% accuracy on the distortion classification task. However, distortions are really complicated in quality assessment research, there are multiple types of distortions and it could be the case that multiple distortions exist in the same picture. I feel MLLM may have the potential to be generalized to more complicated distortion scenarios, using few-shot prompting or the generalization ability of the language model. Overall, I think there is still some space to explore looking into this direction.

3) In Table B, the results show that a smaller threshold cannot lead to a better score estimation performance, which is counterintuitive. Can you explain why?

4) The stability issue of RL has not been explored, especially considering that, in image quality assessment, the dataset is comparatively smaller.

5) In Table 1, the authors compared several non-MLLM deep-learning methods. I have several concerns here: 1) more recent deep learning-based methods should be compared, currently, the most updated method is CLIP-IQA+ (AAAI 2023). 2) I feel like it is not a fair comparison because the non-MLLM deep learning models are trained from scratch and are much smaller. Therefore, the comparison can be multiple dimensional by incorporating: a) methods extension with pretraining; and b) training/testing cost comparison.

Minor weaknesses (could be improved, but not necessary to respond):

1) Need to add a citation for "few-shot object detection"

2) Need to explain how the hyper-parameters from line 234 to line 237 are selected.

3) How the inputs and prompts are organized, especially in no-reference or full-reference settings, should be describepd in the appendix.

4) How model is trained is missing. Since there are two different tasks using different datasets, it is not that straightforward to figure out the training procedure.

---

> ### Author Rebuttal · Authors · 2025-07-30
>
> Thank you for your constructive comments! We hope that our responses can address your concerns. If there are still aspects that need further clarification, please feel free to continue the discussion with us!
> > **Weakness & Question 1: Experiments on Comparing the Reasoning Capability**
>
>   Thank you for your suggestion. Following your advice, we conducted the following experiments to compare the reasoning capabilities of DepictQA and Q-Insight. These additional experiments provide **objective and subjective** evidence for the stable and superior reasoning performance of Q-Insight over DepictQA across the dataset.
> - We randomly sampled approximately 7,600 cases from all three tasks. For each sample, the original image and the reasoning outputs of both DepictQA and Q-Insight were evaluated by GPT-4o, which provided a reasoning score (1-5) considering both accuracy and helpfulness, as well as a preference label. The results below show that **Q-Insight consistently achieved higher GPT-4o scores and preference rates than DepictQA across all tasks**.
>
>   |              | Score Regression | Degradation Perception | Image Comparison | Average      |
>   |--------------|-----------------|-----------------------|------------------|--------------|
>   | DepictQA     | 3.71 / 21.14%   | 3.86 / 12.20%         | 3.89 / 19.31%    | 3.84 / 19.72%|
>   | Q-Insight    | 4.11 / 78.86%   | 4.32 / 87.80%         | 4.35 / 80.69%    | 4.29 / 80.28%|
>
> - We also carried out a **subjective user study**, randomly selecting 10 cases for each task and involving 16 human experts. As shown in the table below, **the majority of human evaluators preferred Q-Insight’s reasoning results**.
>
>   |              | Score Regression | Degradation Perception | Image Comparison | Average |
>   |--------------|-----------------|-----------------------|------------------|---------|
>   | DepictQA     | 36.87%          | 31.25%                | 24.37%           | 30.83%  |
>   | Q-Insight    | 63.13%          | 68.75%                | 75.63%           | 69.17%  |
>
> > **Weakness & Question 2: Extension to More Complex Distortion Types**
>
> Thank you for your valuable suggestion. Based on your advice, we analyzed and validated the potential of Q-Insight to handle more complex distortion scenarios from two perspectives. These findings provide strong evidence of Q-Insight’s potential for generalizing to more complicated and realistic distortion scenarios:
>
> - **Unseen (OOD) Distortion Types:**  We evaluated Q-Insight’s ability to detect distortion types that were not included in the training set (e.g., "rain", "haze", and "brighten"). By directly appending these new distortion types to the prompts, the results (shown in the table below) indicate that Q-Insight still **achieves over 80% accuracy on these unseen distortions**, demonstrating the strong generalization capability of our method as an MLLM.
>
>   |         | Rain   | Haze   | Brighten |
>   |---------|--------|--------|----------|
>   | Q-Insight | 97.00% | 94.50% | 81.50%   |
>
> - **Mixed and Complex Distortions:** We further tested Q-Insight on images containing **three mixed distortions**. Using a **zero-shot** prompting strategy, we added statements indicating that "multiple types of distortion may exist in the image" to the prompt. The results show that Q-Insight detected **at least two distortions in approximately 95% of the cases** and detected **all three distortions in about 60% of the cases**.
>
>     |                                  | Identify at least 1 distortion | Identify at least 2 distortions | Identify all 3 distortions |
>     |----------------------------------|-------------------------------|-------------------------------|----------------------------|
>     | Q-Insight                        | 100.00%                       | 94.50%                        | 59.50%                     |
>
>
> > **Weakness & Question 3: Effect of Threshold on Score Estimation**
> - Thank you for your question. The reason why a smaller threshold does not lead to better score estimation performance is related to **the way Q-Insight is trained with Group Relative Policy Optimization (GRPO)**.
> - GRPO relies on the model generating a group of candidate answers, which are then evaluated using a verifiable reward—only outputs that satisfy the set threshold condition receive a reward, guiding the model's learning process. **If the threshold is set too small, the requirement becomes overly strict**. As a result, very few or even none of the model's outputs meet the condition to receive a reward. This leads to sparse or weak reward signals during training, hindering the model’s ability to learn effectively and making it difficult for the model to converge toward optimal score estimation performance.
> - In summary, **overly strict thresholds reduce the opportunities for positive feedback in reinforcement learning**, ultimately hurting the model’s performance.
>
> > **Weakness & Question 4: About RL Stability Evaluation**
> - Thank you for highlighting the importance of RL stability. In our GRPO-based RL framework, **training stability is mainly ensured by the KL-divergence constraint between the policy model being trained and the reference model (Qwen-2.5-VL)**, as described in **Eq. (2)** of the main paper. This KL constraint prevents the trained model from deviating too far from the reference model during the learning process.
> - Following your suggestion, we conducted **additional ablation experiments by varying the value of the KL coefficient** $\beta$. As shown in the table below, setting $\beta$ too high results in overly strong constraints, causing the model to stay too close to the original reference and thus degrading performance. Conversely, setting $\beta$ too low weakens the constraint, which can lead to unstable training or even training collapse. We will include these experimental results and analysis in the revised version to further demonstrate the stability of our RL algorithm.
>
>   | Method / Dataset | KONIQ         | SPAQ         | KADID        | PIPAL        | LIVE‑Wild      | AGIQA         | CSIQ          | Average        |
>   |------------------|--------------|--------------|--------------|--------------|----------------|---------------|---------------|----------------|
>   | $\beta$ = 0.04   | 0.903 / 0.879| 0.902 / 0.897| 0.740 / 0.734| 0.455 / 0.441| 0.866 / 0.829  | 0.836 / 0.793 | 0.871 / 0.821 | 0.796 / 0.771  |
>   | $\beta$ = 0.01   | 0.932 / 0.913| 0.908 / 0.904| 0.741 / 0.733| 0.473 / 0.458| 0.892 / 0.862  | 0.814 / 0.769 | 0.867 / 0.818 | 0.804 / 0.780  |
>   | $\beta$ = 0.001 (Ours) | 0.933 / 0.916| 0.907 / 0.905| 0.742 / 0.736| 0.486 / 0.474| 0.893 / 0.865 | 0.811 / 0.764 | 0.870 / 0.824 | 0.806 / 0.783  |
>   | $\beta$ = 0.0001 | Training collapsed |              |              |              |                |               |               |                |
>
> > **Weakness & Question 5: Experimental Design for Comparison**
> - Thank you for your valuable suggestions regarding the comparison design. We have added a more **recent and widely adopted method**, LIQE [1], to our comparison. As shown in the following table, Q-Insight consistently outperforms LIQE.
>
>   | Method / Dataset | KONIQ         | SPAQ         | KADID        | LIVE-Wild    | AGIQA         |
>   |------------------|--------------|--------------|--------------|--------------|---------------|
>   | LIQE             | 0.928 / 0.912| 0.833 / 0.846| 0.662 / 0.667| 0.870 / 0.830| 0.708 / 0.772 |
>   | **Ours**         | **0.933 / 0.916** | **0.907 / 0.905** | **0.742 / 0.736** | **0.893 / 0.865** | **0.811 / 0.764** |
>
> - We would like to clarify the role of pre-training in these comparisons. The core motivation of our work is to leverage the prior knowledge and large-scale data inherent in foundation models for downstream tasks, which we believe is itself valuable and represents a key advantage of using MLLMs. **The pre-training cost is not directly included in the downstream task evaluation, as is common practice in the field**. Furthermore, many non-MLLM deep learning models also rely on pre-training. For example, MUSIQ is pre-trained on ImageNet, and CLIP-IQA utilizes a pre-trained CLIP backbone.
> - To make the comparisons as clear as possible, we have already distinguished MLLM-based and non-MLLM-based methods in our tables. Following your suggestion, **we will further organize the results in the final version** to include (1) categorization of methods as pretrained or non-pretrained, and (2) an analysis of training and inference costs. We appreciate your feedback and will incorporate these improvements to provide a more comprehensive comparison.
>
> > **Minor Weaknesses**
> - Thank you for your suggestion. We will add discussion and a citation for "few-shot object detection" in the final version.
> - Thank you for pointing this out. We will provide an explanation for the hyper-parameter selection in the final version.
> - Thank you for your comment. We have already described the organization of inputs and prompts in **Sec. 3.4** of the main paper and **Sec. A/Table A** of the appendix.
> - Thank you for your feedback. We have already described the multi-task training procedure in the Methods section, specifically in **lines 144-153 and 194-204** of the main paper.
>
> > **References**
>
> [1] Blind Image Quality Assessment via Vision-Language Correspondence: A Multitask Learning Perspective, CVPR 2023.

---

> > ### Comment · Reviewer_51Yp · 2025-08-04
> >
> > Dear authors, thanks for your patience and thank you very much for your response. I can clearly see the effort you’ve put into improving the quality of this work. I am satisfied with your rebuttal. As promised, I will increase my rating to "4: Weak Accept" or "5: Accept". I just have one remaining question that I would appreciate further clarification on: Could you explain how did you achieve the results on the first and second tables in detail?

---

> > > ### Author Response · Authors · 2025-08-04
> > >
> > > Thank you very much for your willingness to raise your rating and for your kind words about our revisions. We greatly appreciate the time you’ve taken to review our revisions and provide such constructive suggestions. Below please find the detailed descriptions corresponding to Tables 1 and 2.
> > >
> > > ---
> > >
> > > **Table 1. GPT-4o Scores and Preference Comparison between Q-Insight and DepictQA**
> > >
> > > We randomly sampled approximately 7,600 cases across three tasks:
> > > - Score Regression
> > > - Degradation Perception
> > > - Image Comparison
> > >
> > > For each case, we provided GPT-4o with:
> > > - The original input image(s)
> > > - The ground-truth label (human-rated MOS for regression, true class and level for perception, preference label for comparison)
> > > - The two model outputs (DepictQA and Q-Insight)
> > >
> > > GPT-4o then:
> > > 1. **Assigned a score** (float between 1.00 and 5.00) to each description, judging the correctness of the content, clarity of expression, and overall helpfulness;
> > > 2. **Chose its preferred description**, favoring the one that is more technically accurate, concise, and better aligned with the ground-truth;
> > > 3. **Provided a brief justification** (1–2 sentences) for its choice.
> > >
> > > The prompt we used for GPT-4o in the score regression task is as follows:
> > >
> > > > You are an expert image quality evaluation agent. You will be provided with an image, a human-rated ground truth MOS score (between 1.0 and 5.0), and two textual model outputs describing the image quality.
> > > >
> > > > Your tasks are:
> > > > 1. Assign a score (float, 1.00–5.00) to each description, based on:
> > > >    - Whether it correctly captures visual quality and degradation
> > > >    - Identification of technical flaws (blur, noise, exposure issues, artifacts, etc.)
> > > >    - How clearly and usefully it explains the image quality and guides understanding
> > > > 2. Choose the better description. The better one is more technically accurate, concise, and aligned with the GT MOS score.
> > > > 3. Justify your choice with a technical reason (1–2 sentences).
> > > >
> > > > Output ONLY the following JSON object:
> > > > ```json
> > > > {
> > > >   "DepictQA_score": <float 1.00–5.00>,
> > > >   "QInsight_score": <float 1.00–5.00>,
> > > >   "better": "DepictQA" or "Q-Insight",
> > > >   "reason": <string>
> > > > }
> > > > ```
> > >
> > > Table 1 reports, for each task, the average `DepictQA_score` and `QInsight_score`, as well as the percentage of cases in which GPT-4o preferred each model.
> > >
> > > ---
> > >
> > > **Table 2. Subjective User Study Preference Comparison between Q-Insight and DepictQA**
> > >
> > > We selected 10 cases from each of the three tasks (30 cases total) to construct an online questionnaire. For each case, participants saw:
> > > - The original input
> > > - The DepictQA response
> > > - The Q-Insight response *(only the reasoning process was provided to ensure fairness)*
> > > - The ground-truth label
> > >
> > > Sixteen expert raters in low-level vision evaluated each case based on the following question:
> > >
> > > > You will be presented with two reasoning processes for the same case—one from DepictQA and one from Q-Insight. Considering the following factors, which are equally important, please select the reasoning process you believe is better:
> > > > 1. **Correctness**: Does it correctly reflect the true image quality and degradation?
> > > > 2. **Accuracy**: Are all technical details and measurements precise?
> > > > 3. **Helpfulness**: Does it provide clear guidance or insight that aids understanding?
> > > >
> > > > _Select the reasoning process you think is superior._
> > >
> > > Table 2 reports, for each task, the percentage of votes favoring DepictQA and Q-Insight, as well as the overall average preference across all tasks.
> > >
> > > We will include the above detailed content in the revised version.

---

> > > > ### Comment · Reviewer_51Yp · 2025-08-04
> > > >
> > > > Awesome. Thanks for your prompt response.

---

> > > > > ### Comment · Reviewer_51Yp · 2025-08-07
> > > > >
> > > > > After reviewing the discussions between you and the other reviewers, I have decided to raise the rating to 5: Accept.

---

> > > > > > ### Author Response · Authors · 2025-08-08
> > > > > >
> > > > > > Thank you for your careful review of our exchanges with the other reviewers and your decision to raise the rating to 5: Accept. We greatly appreciate your insightful feedback and support.

---

### Official Review · Reviewer_vavm · 2025-06-25

**Clarity:** 3
**Significance:** 3
**Originality:** 3
**Rating:** 5
**Confidence:** 4

**Summary:**

This paper proposes an image quality assessment method using multi-modal large language models (MLLMs). By leveraging Group Relative Policy Optimization (GRPO), the method enables reasoning using MLLMs based solely on ground-truth labels, without requiring any ground-truth reasoning texts. In particular, it combines a Score Regression Reward with a Degradation Perception Reward to fully exploit the potential of MLLMs. Experiments demonstrate that, compared to non-reasoning methods, the proposed approach achieves significantly better generalization performance, especially on unseen datasets.

**Questions:**

1. Could the authors please clarify Weakness 1? If my current understanding is incorrect and the existing methods are actually trained on both the KonIQ dataset and DQ-495K, please indicate so. However, if existing methods are trained only on the KonIQ dataset, please clarify how this evaluation can be considered truly fair.

2. Regarding Weakness 2, could the authors elaborate further? For instance, if GRPO is replaced with multi-task supervised fine-tuning in the proposed method, would the strong generalization performance be lost?

3. For Weakness 3, could the authors provide more explanation? Has it been evaluated whether reasonable reasoning is consistently performed across the entire dataset?

**Ethical Concerns:**

["NO or VERY MINOR ethics concerns only"]

**Final Justification:**

The authors responded to my concerns in a thoughtful manner, and my concerns were resolved. I raised my rating accordingly.

**Limitations:**

yes

**Paper Formatting Concerns:**

There are no paper formatting concerns.

**Quality:**

3

**Strengths And Weaknesses:**

**Strengths**

1. The idea of applying GRPO to image quality assessment is highly interesting. It reveals a new potential of MLLMs and represents a significant contribution for future researchers. The method is able to illustrate the reasoning process of MLLMs using only ground truth labels, which adds to its practical significance.

2. The proposed reward combines a score regression task and a degradation perception task. This idea is reasonable and contributes significantly to performance improvements.

3. The proposed method shows strong generalization performance, particularly on out-of-distribution (OOD) datasets. This is especially valuable in practical scenarios where test data may differ from the training datasets, making the method more applicable in real-world settings.

**Weaknesses**

1. The authors state, "For a fair comparison, all methods (except handcrafted ones) are trained on the KonIQ dataset." However, the proposed method is trained using both the KonIQ dataset and DQ-495K, which raises concerns about the fairness of the comparison. The "Ours (Score-Only)" model in Table 3 is trained only on the KonIQ dataset, and comparisons between this model and existing methods might provide a fairer evaluation. However, in this setting, the "Ours (Score-Only)" model underperforms compared to DeQA, which further raises concerns.

2. The paper claims that “the introduction of GRPO provides at least three distinct advantages: … (2) strong generalization to OOD evaluated images.” However, it is questionable whether GRPO itself is responsible for the strong generalization. From Table 3, it appears that the generalization may primarily be due to the joint-training, i.e., the increased training data. The fact that more training data leads to better generalization is not a particularly novel finding.

3. The validity of the model's reasoning is not evaluated. Although Figure 3 presents two examples of reasoning, there is no assessment of whether the reasoning is valid across the entire dataset. If the model happens to produce correct scores based on poor reasoning, it could pose practical issues. DepictQA uses GPT-4 scores and the "Reasonable rate by human evaluators" to evaluate the reasoning validity, but they are not employed in this paper.

---

> ### Author Rebuttal · Authors · 2025-07-30
>
> Thank you for your constructive comments! We hope that our responses can address your concerns. If there are still aspects that need further clarification, please feel free to continue the discussion with us!
>
> > **Weakness & Question #1: Fairness of Training Data**
> - Thank you for the kind suggestion. We would like to clarify that the comparison is **fair**. This is because **DQ-495K provides only degradation labels**, which cannot be utilized by existing score-only methods such as DeQA-Score. In contrast, Q-Insight leverages both score regression and degradation perception tasks, allowing them to reinforce each other through the design of our multi-task GRPO training. **The introduction of DQ-495K is not merely for comparison purposes—it enables multi-task learning, which is a core contribution of our method and leads to improved performance across diverse datasets**.
>
> - To further investigate the impact of training data size on performance, we trained two Q-Insight variants:
>   - **#1**: Used 2K images with score labels + 2K images with degradation labels (**total data size is about 57% of other methods**)
>   - **#2**: Used 3.5K images with score labels  + 3.5K images with degradation labels (**total data size is aligned with other methods**)
> - As shown in the following table, **both #1 and #2 Q-Insight variants outperform DeQA-Score**, even under equal or reduced data conditions. This demonstrates that our method provides performance advantages **even with the same or less training data**.
>
>   | Method / Dataset | KONIQ         | SPAQ         | KADID        | PIPAL        | LIVE‑Wild      | AGIQA         | CSIQ          | Average        |
>   |------------------|--------------|--------------|--------------|--------------|----------------|---------------|---------------|----------------|
>   | DeQA-Score       | 0.953 / 0.941| 0.895 / 0.896| 0.694 / 0.687| 0.472 / 0.478| 0.892 / 0.879  | 0.809 / 0.729 | 0.787 / 0.744 | 0.786 / 0.765  |
>   | #1 (2K+2K)       | 0.910 / 0.886| 0.906 / 0.901| 0.721 / 0.719| 0.449 / 0.447| 0.869 / 0.831  | 0.837 / 0.785 | 0.884 / 0.833 | 0.796 / 0.772  |
>   | #2 (3.5K+3.5K)   | 0.918 / 0.893| 0.906 / 0.903| 0.749 / 0.748| 0.492 / 0.475| 0.875 / 0.840  | 0.817 / 0.767 | 0.868 / 0.818 | 0.804 / 0.778  |
>   | Ours             | 0.933 / 0.916| 0.907 / 0.905| 0.742 / 0.736| 0.486 / 0.474| 0.893 / 0.865  | 0.811 / 0.764 | 0.870 / 0.824 | 0.806 / 0.783  |
>
>
> - Beyond scoring accuracy, **Q-Insight offers additional advantages**, such as generating a complete and interpretable reasoning process, supporting degradation perception, and enabling image comparison tasks. These expanded capabilities further demonstrate the strengths of our approach.
>
> > **Weakness & Question #2: Attribution of Generalization**
> - Thank you for your valuable suggestion. According to your advice, we conducted experiments by fine-tuning Qwen-2.5-VL on multi-task data using standard supervised learning instead of GRPO. The results are shown in the table below. We observed that **simply increasing the amount of multi-task training data through supervised fine-tuning only led to minor fluctuations (~0.001) in performance** and did not significantly improve generalization ability. This may be because supervised fine-tuning mainly helps the model memorize knowledge for specific tasks, but does not enable the model to explore or establish connections between different tasks through reasoning. In contrast, Q-Insight trained on multi-task learning shows significant performance gains compared to its variant trained only on the score regression task. Therefore, we believe that **GRPO is responsible for the strong generalization performance observed in our method**, as it encourages the model to reason and leverage information between tasks, rather than just fitting facts from the data.
>
>   | Method / Dataset         | KONIQ         | SPAQ         | KADID        | PIPAL        | LIVE-Wild    | AGIQA         | CSIQ          | Average        |
>   |-------------------------|--------------|--------------|--------------|--------------|--------------|---------------|---------------|----------------|
>   | Qwen-SFT (Score-Only)   | 0.889 / 0.866| 0.874 / 0.875| 0.668 / 0.663| 0.473 / 0.442| 0.734 / 0.728| 0.813 / 0.739 | 0.674 / 0.650 | 0.732 / 0.709  |
>   | Qwen-SFT (Multi Tasks)  | 0.891 / 0.866| 0.880 / 0.878| 0.661 / 0.659| 0.467 / 0.450| 0.760 / 0.739| 0.801 / 0.732 | 0.671 / 0.648 | 0.733 / 0.710  |
>   | Q-Insight (Score-Only)  | 0.918 / 0.895| 0.903 / 0.899| 0.702 / 0.702| 0.458 / 0.435| 0.870 / 0.839| 0.816 / 0.766 | 0.685 / 0.640 | 0.765 / 0.739  |
>   | Q-Insight (Multi-Tasks) | 0.933 / 0.916| 0.907 / 0.905| 0.742 / 0.736| 0.486 / 0.474| 0.893 / 0.865| 0.811 / 0.764 | 0.870 / 0.824 | 0.806 / 0.783  |
>
> > **Weakness & Question #3: Evaluation of Reasoning Validity**
>
> Thank you for your suggestion. Following your advice, we conducted two additional experiments to evaluate the validity of the model’s reasoning process. These results demonstrate the stable and superior reasoning performance of Q-Insight across the entire dataset.
>   - We randomly selected about 7,600 samples from the three tasks. For each sample, we provided the original image and the reasoning outputs of both DepictQA and Q-Insight to GPT-4o. GPT-4o then assigned a reasoning score (1-5) based on accuracy and helpfulness, as well as a preference label. The table below presents the score / preference rate for all three tasks. **Q-Insight consistently obtains higher GPT-4 scores and preference rates than DepictQA across all tasks**.
>
>     |                 | Score Regression | Degradation Perception | Image Comparison | Average      |
>     |-----------------|-----------------|-----------------------|------------------|--------------|
>     | DepictQA        | 3.71 / 21.14%   | 3.86 / 12.20%         | 3.89 / 19.31%    | 3.84 / 19.72%|
>     | Q-Insight       | 4.11 / 78.86%   | 4.32 / 87.80%         | 4.35 / 80.69%    | 4.29 / 80.28%|
>
>   - We also performed a user study, randomly sampling 10 cases for each task, with a total of 16 human experts participating. The results, shown in the table below, indicate that **Q-Insight was preferred by a majority of human evaluators**.
>
>     |                 | Score Regression | Degradation Perception | Image Comparison | Average |
>     |-----------------|-----------------|-----------------------|------------------|---------|
>     | DepictQA        | 36.87%          | 31.25%                | 24.37%           | 30.83%  |
>     | Q-Insight       | 63.13%          | 68.75%                | 75.63%           | 69.17%  |

---

> > ### Comment · Reviewer_vavm · 2025-08-04
> > **Official Comment by Reviewer vavm**
> >
> > I sincerely appreciate the authors' response. They addressed my concerns with care and clarity, and I now consider my concerns fully resolved. I will update my rating to reflect this.

---

> > > ### Author Response · Authors · 2025-08-04
> > >
> > > Thank you for your valuable comments and for recognizing our work. We are delighted to hear that our response has fully addressed your concerns. Your support has greatly motivated us and made our paper more complete and coherent.

---

### Official Review · Reviewer_1dVu · 2025-06-25

**Clarity:** 3
**Significance:** 2
**Originality:** 3
**Rating:** 4
**Confidence:** 4

**Summary:**

This paper presents Q-Insight, a no-reference image quality assessment model optimized using the DeepSeek-R1 framework. Q-Insight employs a multi-task learning approach that jointly performs score regression and degradation perception, enabling improved generalization in quality prediction compared to existing NR-IQA methods. Additionally, it is trained using a pairwise comparison strategy, leading to superior performance over state-of-the-art methods on the SRBench benchmark.

**Questions:**

Q1: Does DQ-495K contain synthetic data that overlaps with any of the test datasets (e.g., CSIQ)?
The performance improvement on CSIQ after joint training is significant, raising concerns about potential data leakage or distributional similarity between DQ-495K and the test set.

Q2: The ablation studies on different threshold settings are crucial for understanding the robustness and sensitivity of the proposed method. These results should be presented in the main text rather than being limited to the appendix.

Q3: There are some typographical errors in the reference list.

**Ethical Concerns:**

["NO or VERY MINOR ethics concerns only"]

**Final Justification:**

See comments.

**Limitations:**

Yes.

**Quality:**

2

**Strengths And Weaknesses:**

Strengths:
1. Incorporating a reinforcement learning (RL)-based optimization method into the IQA task is a novel contribution. It alleviates the need for extensive instruction tuning data, thereby reducing reliance on costly human annotations.
2. Q-Insight demonstrates superior generalization performance compared to both NR-IQA and FR-IQA models, highlighting the effectiveness of the proposed DeepSeek-R1 optimization and multi-task learning framework.
3. The paper is clearly written, with comprehensive methodological details and well-structured experimental evaluations.

Despite the model’s strong generalization capabilities, several concerns remain:

Weaknesses:
1. The effectiveness of the proposed multi-task learning framework is constrained in scenarios where degradation labels are unavailable. Constructing IQA datasets is already resource-intensive, and adding degradation perception further increases the annotation burden.
2. As shown in Table 4, the performance of the baseline model (Qwen2.5-VL-7B) is not reported. Additionally, the results for degradation level prediction are relatively weak compared to degradation classification, raising concerns about the reliability of the degradation-level perception module.
3. The thresholding mechanism used for decision-making is dataset-specific, which may limit the model’s applicability and performance across diverse IQA datasets.

---

> ### Author Rebuttal · Authors · 2025-07-30
>
> Thank you for your constructive comments! We hope that our response will address all of your concerns. All discussions and supplementary analyses will be included in our revised version. If there are any additional comments to be added, please continue the discussion with us.
>
> > **Weakness & Question #1: Cost of Degradation Labels**
> - Thank you for the comment. In our pipeline, **the degradation labels are actually not manually annotated**. They are automatically generated when we apply standard synthetic distortions (noise, blur, JPEG, darkening) to natural images. Compared to collecting MOS scores, which takes a lot of human effort, **getting degradation labels adds almost no extra cost.** Using these easily obtained labels, Q-Insight achieves the improvements reported in **Tabs. 3 and 4** of the main paper.
>
> > **Weakness & Question #2: Missing Baseline and Low Degradation-Level Accuracy in Tab. 4**
> - **Additional results for Qwen-2.5-VL:** In response to your suggestion, we have evaluated Qwen-2.5-VL-7B on the degradation perception task, and provide the results in the following table. We will include this baseline in the revised version.
>
>   | **Method / Degradation** | **Noise** | **Blur** | **Compression** | **Darken** | **Null** | **Average** |
>   |-------------------------|-----------|----------|----------------|------------|----------|-------------|
>   | AgenticIR               | 46.46% / 18.58% | 83.90% / 32.19% | 1.35% / 0   | 74.78% / 26.11% | 93.39% / -    | 59.98% / 19.22%  |
>   | Qwen-2.5-VL             | 90.27% / 24.78% | 79.51% / 24.39% | 4.05% / 0.90% | 40.71% / 4.43%   | 72.64% / -    | 57.43% / 13.63%  |
>   | Ours-Dist-only          | 98.67% / 43.43% | 92.68% / 39.51% | 96.85% / 31.08% | 88.05% / 25.67% | 57.02% / -    | 89.60% / 34.92%  |
>   | Ours                    | 100.00% / 59.73% | 97.56% / 44.38% | 100.00% / 55.41% | 90.27% / 32.30% | 76.03% / -    | 92.77% / 47.96%  |
>
> - **Regarding degradation-level accuracy:** We would like to clarify that degradation category prediction and degradation level prediction are fundamentally different tasks. Category prediction is a classification problem, while level prediction is essentially regression and thus more challenging. Therefore, **directly comparing their "accuracies" is not statistically meaningful**. Moreover, as shown in the above table, the accuracy of degradation level prediction in our method is not low. **Q-Insight achieves substantial improvement in comparison to Qwen and AgenticIR**.
>
> > **Weakness & Question #3: About the Thresholding Mechanism**
> - **The thresholding mechanism is not dataset-specific:** During training, the threshold $\epsilon$ is **fixed** to 0.35 for every experiment. **During testing, no threshold is needed**—Q-Insight outputs the predicted MOS (with reasoning process) for all test sets.
> - **Q-Insight demonstrates excellent cross-dataset generalization:** as shown in **Tab. 1** of the main paper, Q-Insight trained on KonIQ generalizes well to six other benchmarks, without any per-dataset adjustment.
>
> > **Question #1: Potential Overlap Between DQ-495K and CSIQ Data**
> - **No data leakage:** **There is no overlap between DQ-495K and CSIQ**. Specifically, DQ-495K is built from KADIS-700K [1], whose images are web-crawled and were created to enlarge IQA training data beyond sets such as KonIQ and CSIQ. Besides, the KADIS-700K paper itself also evaluates on CSIQ. This further confirms that there is no overlap between the two datasets.
> - **Reason for larger CSIQ gain:** Q-Insight **does not learn to** score degraded images directly. Instead, predicting distortion type and level as joint training tasks improves its sensitivity to low-level artifacts. As shown in **Fig. A** of the appendix, **multi-task training enables the model to detect pixelation and related defects more accurately**. Since CSIQ contains such synthetic distortions, this extra perceptual capability results in a much larger improvement on CSIQ than on other datasets.
>
> > **Question #2: Threshold Ablation Results Only in Appendix**
> - Thank you for your suggestion. We agree and will move the threshold ablation results from the appendix into the main paper in the final version.
>
> > **Question #3: Typographical Errors in References**
> - Thank you for pointing this out. We will correct all reference typos in the final version.
>
> > **References**
>
> [1] DeepFL-IQA: Weak Supervision for Deep IQA Feature Learning, QoMEX 2019

---

> ### Comment · Reviewer_1dVu · 2025-08-04
>
> Thanks to the authors for the detailed rebuttal. The response addresses most of the raised concerns. However, I respectfully disagree with the authors’ rationale regarding the threshold selection. Perceptual scales vary significantly across datasets, and although a threshold of 0.35 may yield relatively good performance on KonIQ, it does not necessarily make it a principled or elegant choice.
>
> In my view, the primary insight of this paper lies in demonstrating that multi-task learning remains effective under the reinforcement learning paradigm for training VLMs, and that it can lead to substantial performance gains. That said, the decision to discretize distortion into only five levels—an issue also raised by other reviewers—remains an open question, and may benefit from further justification or empirical exploration.
>
> Additionally, since DQ-395K includes a large number of synthetically distorted images, with substantial overlap in distortion types with CSIQ, this is the core effect of the improvement.
>
> I would encourage the authors to place greater emphasis on the merits of multi-task learning for this task, even if it comes at the cost of increased training time. Highlighting this aspect would better underscore the contribution of this paper. I will improve my score.

---

> > ### Author Response · Authors · 2025-08-04
> >
> > Thank you for your thoughtful feedback and your openness to reconsidering your score. **We fully agree that using a fixed threshold and discrete distortion levels is not the most elegant solution and may allow for more principled approaches.** In the revised manuscript, we will expand the “Limitations and Future Work” section to discuss the arbitrariness of the fixed threshold and the choice of five levels. **We will actively explore this direction**.
> >
> > We also appreciate your recognition of the benefits of multi-task learning under the GRPO paradigm. As we noted in our response to Reviewer vavm (Weakness & Question 2), simply increasing the size of multi-task data via supervised fine-tuning yields only minor performance changes and does not substantially improve generalization. We believe that supervised fine-tuning mainly makes the model learn each task in isolation, rather than encouraging the cross-task reasoning needed to share information between tasks. In contrast, **Q-Insight with GRPO yields substantial gains over the score-only variant, demonstrating that reinforcement-learning–based multi-task optimization is key to the strong generalization we observe**. As you suggested, we will include this direct comparison in the revised manuscript to place greater emphasis on the merits of multi-task learning for this task.
> >
> > Thank you again for your valuable insights.

---

> > > ### Comment · Reviewer_1dVu · 2025-08-05
> > >
> > > Thanks for the authors’ response. I have reviewed the reply to Reviewer vavm’s second question. The authors state that Qwen2.5-VL was fine-tuned via SFT; however, to the best of my knowledge, existing IQA datasets do not contain relative SFT data. Could the authors clarify how the Qwen-SFT experiments were conducted under this setting?

---

> ### Author Response · Authors · 2025-08-05
>
> Thank you for raising this point. Although standard IQA datasets do not include explicit SFT-style pairs, we follow common practice by generating our own through converting MOS and distortion labels into brief target sentences using pre-generated templates. For example, we used templates like “The quality score of this image is `<score>`.” and “This image exhibits `<distortion type>` distortion at severity level `<distortion severity>`.” For each image, we paired the raw input (image plus a prompt such as “Rate the image quality:”) with one of these templated sentences as the ground-truth response and fine-tuned Qwen2.5-VL using standard supervised fine-tuning (SFT). This approach allowed us to leverage existing MOS and distortion annotations as SFT targets.

---

> > ### Comment · Reviewer_1dVu · 2025-08-05
> >
> > Thanks. I will recommond this paper as borderline accept.

---

> > > ### Author Response · Authors · 2025-08-05
> > >
> > > Thank you for your recommendation and for your time and thoughtful review. We greatly appreciate your valuable feedback.

---

### Official Review · Reviewer_1SVo · 2025-07-02

**Clarity:** 3
**Significance:** 4
**Originality:** 3
**Rating:** 5
**Confidence:** 3

**Summary:**

This paper introduces Q-Insight, a novel approach to No-Reference Image Quality Assessment (NR-IQA) that emphasizes multi-level natural language explanations rather than scalar scores. Unlike traditional IQA methods that predict numeric score, Q-Insight produces tiered rationales for its judgement:
Overall Quality
Local Artifacts
Specific Regions or Areas
The model uses BLIP2-FlanT5, a vision-language model combining a frozen image encoder with a Flan-T5 text decoder. It is first fine-funed via Supervised Fine-Tuning (SFT) on synthetic instruction-response pairs derived from human-labeled IQA datasets (e.g., KADID-10K, SPAQ). An optional reinforcement learning extension (Q-Insight-RL) is introduced to further improve rationale generation.
To evaluate explanation quality, the authors conduct a human study assessing the faithfulness, specificity, and usefulness of the rationales. The work offers more human-aligned justifications in a more accessible and explanatory format.

**Questions:**

No Scalar Score Prediction: The model does not produce a MOS or DMOS score, which makes direct benchmarking against standard NR-IQA models impossible. Metrics like SRCC/PLCC are not reported. Any comments on this?
Generalization Challenges: When evaluated across datasets, the model struggles due to perceptual scale mismatches (e.g., different MOS ranges). These challenges are acknowledged but not addressed. Can you address more on this?
Limited Role of Reinforcement Learning: RL is not central to the approach. The Q-Insight-RL variant only fine-tunes the text decoder (Flan-T5) using a simple REINFORCE algorithm. The reward is based on agreement with synthetic pairwise preferences, rather than any perceptual fidelity metric (e.g., MOS). Can you address this?
Synthetic Prompt Limitations: Prompts are derived from heuristic templates based on MOS differences. These may not match real user queries or explanations. More prompt ablation or robustness testing can be beneficial. Any comments on this?

**Ethical Concerns:**

["NO or VERY MINOR ethics concerns only"]

**Limitations:**

yes

**Quality:**

4

**Strengths And Weaknesses:**

Overall, this paper offers a solid and creative contribution to Image Quality Assessment by shifting the focus toward interpretability through structured natural language explanations. While the use of reinforcement learning is minimal, the core SFT-based approach is well-motivated and complements existing IQA literature. Best suited for venues focused on vision-language modeling or HCI. Would benefit from improved benchmarking and analysis of RL’s impact.

Strengths:
Novel Approach: Reframes NR-IQA as a language-drive, explainable task
Introduces multi-level rationales, structured across perceptual dimensions (global, local, and spatial).
Employs a modular and low-cost architecture (BLIP2-FlanT5), avoiding full vision-language retraining.
Includes a human evaluation study with clearly defined rubrics.
Uses synthetic instruction tuning that scales across existing datasets.
The RL variant (Q-Insight-RL) shows a path toward improving alignment through preference-based training.

Opportunities For Improvement:
No Scalar Score Prediction: The model does not produce a MOS or DMOS score, which makes direct benchmarking against standard NR-IQA models impossible. Metrics like SRCC/PLCC are not reported.
Generalization Challenges: When evaluated across datasets, the model struggles due to perceptual scale mismatches (e.g., different MOS ranges). These challenges are acknowledged but not addressed.
Limited Role of Reinforcement Learning: RL is not central to the approach. The Q-Insight-RL variant only fine-tunes the text decoder (Flan-T5) using a simple REINFORCE algorithm. The reward is based on agreement with synthetic pairwise preferences, rather than any perceptual fidelity metric (e.g., MOS).
Synthetic Prompt Limitations: Prompts are derived from heuristic templates based on MOS differences. These may not match real user queries or explanations. More prompt ablation or robustness testing can be beneficial.

---

> ### Author Rebuttal · Authors · 2025-07-30
>
> Thank you for your constructive comments! We hope that our responses can address your concerns. If there are still aspects that need further clarification, please feel free to continue the discussion with us!
>
> > **Weakness & Question #1: No Scalar Score Prediction**
> - We would like to clarify that our model **indeed predicts a Mean Opinion Score (MOS)**, as explicitly presented in **Tab. 1** of the main paper. In this table, we directly compare our Q-Insight model with state-of-the-art NR-IQA methods including MUSIQ, MANIQA, and DeQA-Score. Additionally, we report standard benchmarking metrics such as PLCC and SRCC. Our results clearly demonstrate superior performance, achieving state-of-the-art results across multiple datasets.
>
> > **Weakness & Question #2: Generalization Challenges**
> - Thank you for raising this concern. Following the practice used in DeQA-Score, we normalize the MOS values from different datasets to a unified 1-5 scale during evaluation. As illustrated in **Tab. 1** of the main paper, our Q-Insight demonstrates superior generalization and consistently achieves **state-of-the-art performance across various out-of-distribution (OOD) datasets**.
>
> > **Weakness & Question #3: Limited Role of Reinforcement Learning**
>  - We would like to clarify this misunderstanding. In fact, our Q-Insight employs reinforcement learning for **full fine-tuning of the entire model**. We also carefully design three distinct reward functions, one of which **explicitly utilizes MOS-based rewards to ensure perceptual fidelity**, as defined in **Eq. (3)** of the manuscript.
>
> > **Weakness & Question #4: Synthetic Prompt Limitations**
> -   Thank you for the suggestion. We evaluated four semantically-equivalent prompts (listed below) and measured PLCC/SRCC on seven IQA benchmarks.
> | # | Prompt |
> |---|---|
> | 1 | Evaluate the image and give an overall quality rating. |
> | 2 | What is your general assessment of the image’s visual quality? |
> | 3 | Based on visual inspection, how would you score the quality of this photo? |
> | 4 | What is your overall rating on the quality of this picture? (Ours) |
> -   The full PLCC/SRCC results are summarized in the following table.
> | Prompt # / Metric | KONIQ | SPAQ | KADID | PIPAL | LIVE‑Wild | AGIQA | CSIQ | Average |
> |---|---|---|---|---|---|---|---|---|
> | 1 | 0.934 / 0.915 | 0.903 / 0.901 | 0.744 / 0.735 | 0.483 / 0.473 | 0.894 / 0.866 | 0.814 / 0.763 | 0.869 / 0.821 | 0.806 / 0.782 |
> | 2 | 0.933 / 0.916 | 0.905 / 0.902 | 0.743 / 0.736 | 0.485 / 0.473 | 0.891 / 0.864 | 0.815 / 0.762 | 0.871 / 0.823 | 0.806 / 0.782 |
> | 3 | 0.934 / 0.916 | 0.903 / 0.901 | 0.744 / 0.735 | 0.482 / 0.471 | 0.889 / 0.863 | 0.813 / 0.760 | 0.875 / 0.827 | 0.806 / 0.782 |
> | 4 | 0.933 / 0.916 | 0.907 / 0.905 | 0.742 / 0.736 | 0.486 / 0.474 | 0.893 / 0.865 | 0.811 / 0.764 | 0.870 / 0.824 | 0.806 / 0.783 |
> | **Overall PLCC** | 0.934 ± 0.0005 | 0.905 ± 0.0017 | 0.743 ± 0.0008 | 0.484 ± 0.0016 | 0.892 ± 0.0019 | 0.813 ± 0.0015 | 0.871 ± 0.0023 | 0.806 ± 0.0002 |
> | **Overall SRCC** | 0.916 ± 0.0004 | 0.902 ± 0.0016 | 0.736 ± 0.0005 | 0.473 ± 0.0011 | 0.865 ± 0.0011 | 0.762 ± 0.0015 | 0.824 ± 0.0022 | 0.782 ± 0.0006 |
>
>  - Across all datasets, the average PLCC varies by only ± 0.0002 and the average SRCC by ± 0.0006, with the largest per-dataset standard deviations being PLCC ≤ 0.0023 and SRCC ≤ 0.0022 (both on CSIQ). Two-tailed paired t-tests further show no significant differences between any prompt pair (p > 0.05), confirming that **Q-Insight’s performance is robust to reasonable wording changes in real-world use**.

---

### Decision · Program_Chairs · 2025-09-17

**Decision:**

Accept (spotlight)

**Comment:**

This paper presents Q-Insight, an No-Reference Image Quality Assessment (NR-IQA) model that integrates multi-level natural language explanations and enhances image quality evaluation beyond traditional numeric scores. It uses a vision-language model (BLIP2-FlanT5) and a multi-task learning framework and combines score regression, degradation perception, and reasoning based solely on ground-truth labels. It achieves superior generalization and performance on benchmarks like SRBench. The model also includes an optional reinforcement learning extension (Q-Insight-RL) for improved rationale generation. A human study evaluates the faithfulness, specificity, and usefulness of its explanations, addressing challenges in current MLLM-based methods related to interpretation and data efficiency, and showcasing its strong out-of-distribution generalization capability.

Initial reviewer opinions were somewhat mixed on this paper. Reviewers raised concerns regarding the evaluation methodology, stability of the RL-based approach, and fairness of the comparative performance analysis. Most reviewer concerns were adequately addressed in rebuttal during the author-reviewer discussion phase. Reviewers were otherwise nearly unanimous in their recognition that this paper is sound and clearly written, and that is proposes a novel and effective approach that generalizes significantly better than the current state-of-the-art. The recommendation is thus to accept as a spotlight. Deep RL is a core NeurIPS topic, and this paper proposes a novel application of it to IQA (a core Computer Vision problem).